# Genesis and Fluid Evolution of the Hongqiling Sn-W Polymetallic Deposit in Hunan, South China: Constraints from Geology, Fluid Inclusion, and Stable Isotopes

Wenqi Ren [1] , Lei Wang [1,*], Shenjin Guan [1,*] , Jiajin Xu [1] , Hao He [1,2] and Enyi Zhu [1]

1   Faculty of Land Resource Engineering, Kunming University of Science and Technology, Kunming 650031, China
2   Geophysical Exploration Academy of China Metallurgical Geology Bureau, Baoding 071051, China
*   Correspondence: kust_wanglei@kust.edu.cn (L.W.); guansj@kust.edu.cn (S.G.)

**Abstract:** The Hongqiling is a vein-type Sn-W polymetallic deposit in southern Hunan (South China). It is geologically located on the northern margin of the Nanling metallogenic belt. Based on the mineral assemblage and vein crosscutting relationship, three mineralization stages were identified: Sn-W mineralization (S1: cassiterite, wolframite, scheelite, arsenopyrite, molybdenite, pyrite, chalcopyrite, and quartz), Pb-Zn mineralization (S2: chalcopyrite, pyrrhotite, galena, sphalerite, pyrite, quartz, and fluorite), and late mineralization (S3: quartz, fluorite, calcite, galena, sphalerite, and pyrite). According to laser Raman probe analysis, $H_2O$ dominates the fluid inclusions in the S1 and S2 stage quartz, with $CO_2$ and trace $N_2$ following close behind. The ore fluid has low salinity, low density, and a wide temperature range, as per our microthermometric data: the S1 stage has homogenization temperatures (Th) of 236–377.6 °C (average 305.3 °C) and salinity of 3.5–10.7 wt.% NaCleqv; the S2 stage has Th of 206.5–332 °C (average 280.7 °C) and salinity of 1.6–5.1 wt.% NaCleqv; and the S3 stage has Th of 170.9–328.7 °C (average 246 °C) and salinity of 0.2–5.9 wt.% NaCleqv. Based on the results of the aforementioned investigation, the fluid inclusions in quartz, fluorite, and calcite are mainly $H_2O$-NaCl vapor-liquid two-phase. Additionally, examinations of inclusions in S1 wolframite and coexisting quartz using infrared and microthermometry show that the mineralizing fluid likewise belongs to the NaCl-$H_2O$ system. The Th of inclusions in wolframite is ~40 °C higher than that of coexisting quartz. Moreover, the fluid experienced a decrease in temperature accompanied by nearly constant salinity, which indicates that wolframite precipitation is due to fluid mixing and simple cooling, and the precipitation is earlier than quartz. In situ S and H-O isotope data show that the samples have $\delta^{34}S$ = −2.58‰ to 1.84‰, and the ore fluids have $\delta D$ = −76.6 to −51.5‰ (S1 and S2), and $\delta^{18}O_{fluid}$ = −6.6 to −0.9‰ (S1) and −12.9 to −10.2‰ (S2). All these indicate that the mineralizing fluid was derived from the granitic magma at Qianlishan, with substantial meteoric water incursion during the ore stage. Such fluid mixing and subsequent cooling are most likely the primary controls for ore deposition.

**Keywords:** southern Hunan; fluid inclusions; quartz-vein-type wolframite; H-O-S isotopes; fluid mixing



## 1. Introduction

The Nanling belt in south China is one of the world's largest W-Sn provinces [1], it is famous for its large-scale and multi-stage magmatism and abundant W, Sn, and other rare-metal resources and reserves [2–9]. Most tungsten deposits in this region have a close genetic relationship to the granitic magmatism during the Late Mesozoic [10–12]. The Dongpo orefield is located on the northern margin of the middle Nanling belt. It is a typical example of the W-Sn polymetallic mineralization area in Nanling, with rich mineral resources in the field, which contains more than 20 large- to super-large-scale deposits. The Hongqiling Sn-W polymetallic deposit, located on the northern margin of the middle Nanling belt and the outer contact zone on the northeast side of the Dongpo orefield, is

the largest vein-type tin deposit discovered in Hunan Province [13,14]. It consists of the Congshuban Pb-Zn, the Fanlongdui Sn-W-Pb-Zn, and the Hongqiling Sn-W ore sections.

Many studies have been carried out on the Hongqiling deposit since its discovery in 1965, including geological features, age, geochemistry, and metallogeny. Jiang (1987) was the first to investigate the trace and REE compositions of the Qianlishan pluton [13]. Ma et al. (2010) suggested that the geochemistry of Hongqiling is very similar to that of the second-phase intrusive rocks of the Qianlishan pluton based on fluid inclusions Rb-Sr dating of quartz [15]. Meanwhile, Chen et al. (1999) revealed that mineralization occurred at Congshuban at medium-low temperatures and that the ore fluids were magma-derived [16]. Su (2007) concluded that the mineralization of Hongqiling is multistage based on fluid inclusion microthermometry of the various ore veins and muscovite Ar-Ar dating [17]. On the other hand, thorough petrographic investigations on the Sn-W-Pb-Zn ore zoning, as well as analyses on the fluid evolution and metal transport and precipitation mechanism (particularly for wolframite), are still absent due to the limitations of the analysis and measuring methods in the early years. With the recent advancement of in situ mineral analytical technology, including cathodoluminescence (CL) imaging, infrared microscopy, and LA-ICP-MS fluid inclusion analysis, the fluid evolution and mineralization processes in W-Sn deposits can be better elucidated.

In this study, we have thoroughly investigated the fluid inclusions in quartz, wolframite, and late fluorite of the Hongqiling deposit using microthermometry, infrared technology, and laser Raman spectroscopy. We combine this study with in situ S and H-O isotopic compositional characterization to first fill the research gap for the characterization of ore-forming fluids in the study area, and to further understand the fluid evolution process and ore precipitation mechanism. This study also seeks to deepen prospecting prediction and expand regional metallogenic theory, offering fresh perspectives on mineralization mechanisms and ore fluid studies of similar quartz-vein-type Sn-W polymetallic deposits worldwide. At the same time, we hope to provide a new possibility for the mineralization theory of wolframite.

## 2. Geological Setting

### 2.1. Regional Geology

The South China block was formed by the Neoproterozoic amalgamation of the Yangtze and Cathaysia blocks (Figure 1a) [18]. The multiphase tectonic-hydrothermal superposition and diverse mineralization had formed large clusters of ore deposits in South China. The Nanling metallogenic belt is located in central South China and has undergone several tectonic events since the Proterozoic. The final Middle Triassic Yangzi-Cathaysia collision has made Nanling the most important W-Sn polymetallic belt in China and the world [19–21].

The southern Hunan district is located in the Yangzi-Cathaysia suture zone and the overlapping area of the Nanling metallogenic belt and Qinhang metallogenic belt (Figure 1b) [22,23]. The district hosts many nonferrous metal deposits [24,25]. The world-class Dongpo W-Sn-Pb-Zn polymetallic orefield in southern Hunan is well known for its large W-Sn polymetallic deposits [24]. Among these are the Shizhuyuan ultra-large W-Sn-Mo-Bi deposit [26,27], the Jinchuantang Sn-Bi deposit [28], the Yejiwei Sn-Cu deposit [29], and the Nanfengao Pass Pb-Zn-Ag deposit [30]. Hongqiling has received a lot of attention due to the development of the most prominent vein Sn deposit in Hunan Province.

The region has undergone three tectonic cycles in the Caledonian (Early Paleozoic), Hercynian-Indosinian (Late Paleozoic), and Yanshanian (Jurassic-Cretaceous), which developed a series of NE-, NNE-, and EW-trending faults and folds [31]. Upper Proterozoic to Quaternary (except for the Ordovician-Silurian) sedimentary strata are well developed in the region. Many Middle and Late Jurassic plutons are represented by the Qianlishan granite (Figure 1c) [32]. The pluton (exposed area: 10 km$^2$) trended N-S and intruded into the Sinian system's middle-upper series and the Middle-Upper Devonian sequence [33]. Mao et al. (1995) divided this intrusive complex into three generations [26]: (1) pre-ore por-

phyritic biotite granite (152 ± 9 Ma) in the south; (2) syn-ore, equigranular biotite granite that formed the central part of the pluton (137 ± 7 Ma and 136 ± 6 Ma [18]); (3) post-ore granite porphyry (131 ± 1 Ma). Shu et al. (2011) again tested the zircon U-Pb ages of all granite bodies in the west of the Nanling Range. The latest results show that the two early stages of the Qianlishan pluton were formed at 155 ± 2 Ma, and the Mesozoic granitoids in western Nanling Range crystallized in the period from 160 Ma to 150 Ma [34]. The Qianlishan pluton has a significant influence on regional mineralization, and based on the age of Qianlishan pluton, we believe that the vein deposits in the area may have a genetic relationship with the main parts of the Qianlishan pluton. Additionally, it has the potential significance of the formation of the ore [13]. With the Qianlishan pluton as the center, the ore bodies are primarily produced in and around the rock contact zone (Figure 1d) [35]. Due to the development of multiple phases of magmatic-hydrothermal activity and tectonic movement, the fracture structure connected with the intrusive rock becomes a high-quality channel for fluid upward mineralization. Hongqiling Sn-W polymetallic ore is produced in such fracture zone [36].

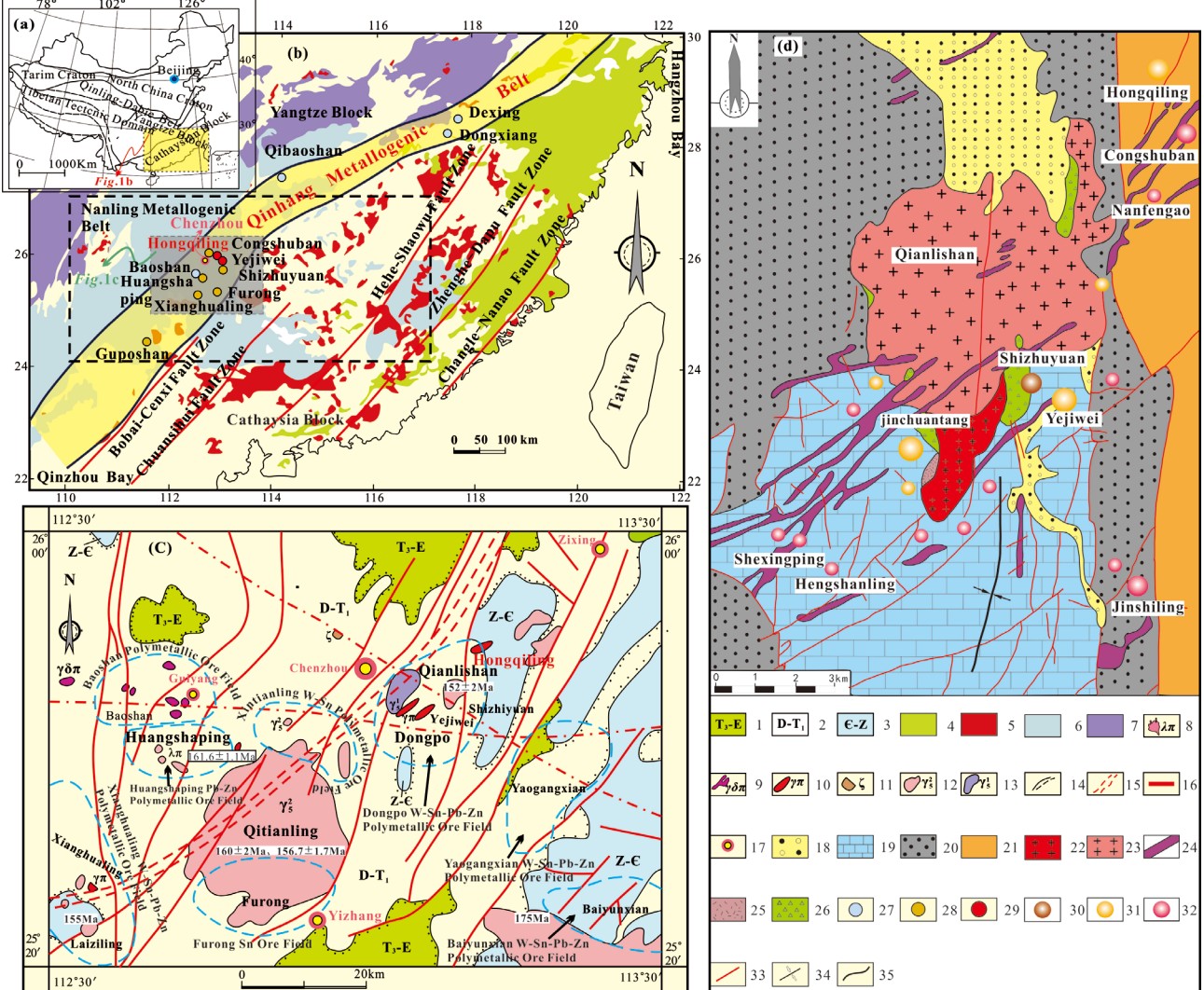

**Figure 1.** (**a**) Location map of the Nanling area; (**b**) distribution of granites and major W-Sn-Cu deposits in the Qinhang and Nanling metallogenic belts (after [37–39]); (**c**) geological sketch of the Qianlishan-Qitianling area (after [40]); (**d**) geological sketch of the Qianlishan granite and surrounding deposits (after [35]). 1. Upper Triassic-Paleogene System; 2. Devonian-Lower Triassic System; 3. Cambrian-Sinian

System; 4. Cretaceous extrusive rock; 5. Jurassic intrusive rock; 6. Jurassic extrusive rock; 7. Jiangnan uplift basement; 8. Quartz porphyry; 9. Granodiorite porphyry; 10. Granite porphyry; 11. Syenite; 12. Early Yanshanian granite; 13. Indsinian granite; 14. Geological boundary; 15. Gravity and magnetic inference of deep fault zones; 16. Regional large fault; 17. City and country; 18. Quaternary sediments; 19. Middle-Upper Devonian carbonate rocks; 20. Middle Devonian Tiaomajian Formation sandstone; 21. Sinian metamorphic clastic rock; 22. Yanshanian porphyritic biotite granite; 23. Yanshanian equigranular biotite granite; 24. Yanshanian granite porphyry; 25. Greisen; 26. Skarn; 27. Middle Jurassic Cu-Au-Ag deposit; 28. Late Jurassic Sn-W deposit; 29. Late Jurassic Pb-Zn deposit; 30. W-Sn polymetallic deposit; 31. Sn polymetallic deposit; 32. Pb-Zn deposit; 33. Fault; 34. Syncline; 35. Geological boundary.

### 2.2. Deposit Geology

The Hongqiling Sn-W polymetallic deposit is located on the NE part of the Qianlishan granite. Exposed stratigraphy includes mainly the Lower Sinian Sizhoushan Formation low-grade meta-sandstone/siltstone and slate, which is the primary ore host [41]. Major faults are NNE-, NE-, and NW-trending, among which the NNE-trending (e.g., F4, F101, F102, F103) and NE-trending (e.g., F3, F27) ones are better developed and closely ore related (Figure 2). The area is characterized by multiphase magmatism, including concealed porphyritic biotite granite ($153 \pm 3$ Ma [42]), equigranular biotite granite ($151 \pm 3$ Ma [42,43]), and granite porphyry-quartz porphyry with diabase in southeastern Hongqiling [14,26]. He (2022) reported cassiterite U-Pb ages of $158.4 \pm 0.8$ Ma and $158.9 \pm 0.7$ Ma, which were interpreted as the mineralization age at Hongqiling [44]. This age is in line with the diagenetic and mineralization ages of many deposits in the Dongpo orefield and the main parts of the Qianlishan pluton, which are 160–150 Ma. This indicates that the mineralizations of ore deposits in this area are also spatially and temporally related to the Qianlishan granite pluton.

There are 52 ore veins at Hongqiling. The general strike of ore veins trends NNE-NE, with the No. 4 being the largest. Additionally, there are two main types of W-Sn mineralization: cassiterite-sulfide type (e.g., No. 4 ore vein) and cassiterite-quartz vein-type (e.g., No. 101, 102, 103 ore veins). The No. 4 ore vein is characterized by W- and Sn-dominated mineralization in the lower and upper parts, respectively, accompanied by minor Pb-Cu-Ag mineralization. The wall rock alterations are mainly silicification and chloritization with an overall vein shape. Ore minerals are predominated by massive textures and are composed of cassiterite, wolframite, pyrrhotite, and chalcopyrite. Ore textures and structures include euhedral-hypidiomorphic granular, cataclastic, and disseminated.

The No. 101, 102, and 103 ore veins occur mainly as veins/veinlets. This group of ore veins is roughly parallel, closely coexisting, and intersects with the No. 4 ore vein in the west. The wall rock alteration occurs through fluoritization. Ore minerals include wolframite, cassiterite, arsenopyrite, scheelite, molybdenite, and chalcopyrite. Ore textures and structures include common crystallization, metasomatic-relict, euhedral-hypidiomorphic granular, banded, massive, and veined.

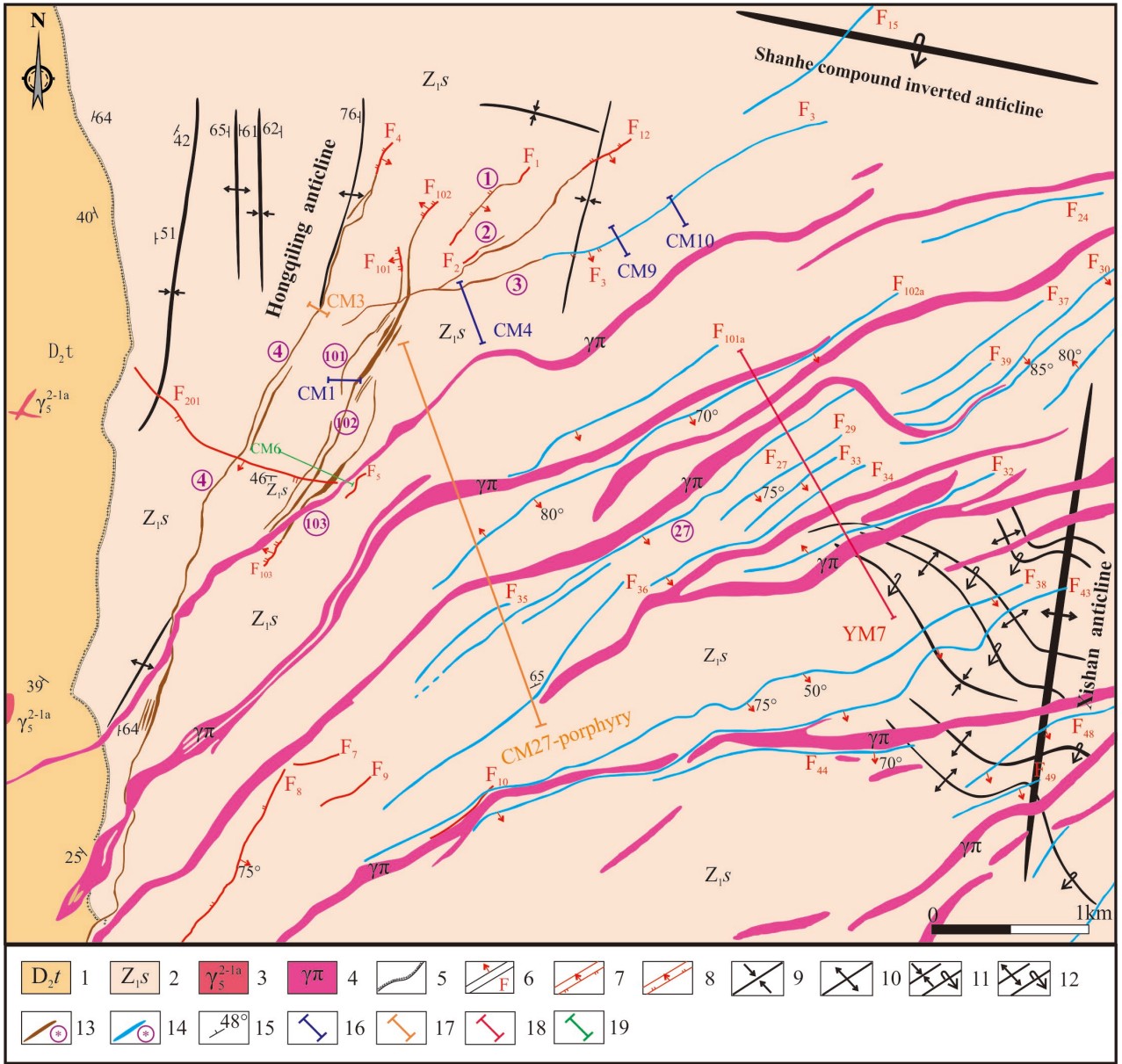

**Figure 2.** Geological sketch of Hongqiling mining area (after [45]). 1. Middle Devonian Tiaomajian Formation; 2. Lower Sinian Sizhoushan Formation; 3. The first intrusive marginal phase of Early Yanshanian granite (2nd member of Qianlishan Formation); 4. Granite porphyry; 5. Unconformity; 6. Fault and number; 7. (inferred) reverse fault; 8. (inferred) normal fault; 9. Syncline; 10. Anticline; 11. EW-trending secondary syncline and inverted anticline; 12. EW-trending secondary anticline and inverted oblique; 13. Sn veins and their numbers; 14. Pb-Zn veins and their numbers; 15. Stratigraphic occurrence; 16. 836 m mine level; 17. 766 m mine level; 18. 600 m mine level; 19. 420 m mine level.

The Pb-Zn ore veins are represented by the ore vein No. 27. This vein is situated on the Congshuban ore section's northeastern edge. Deep lead-zinc mineralization is continuous, and the alteration of No. 27 on the surface is mainly silicification and chloritization. Sphalerite and chalcopyrite are the two most common ore minerals, followed by galena and pyrite. Ore textures and structures include metasomatic, allotriomorphic granular, idiomorphic granular, massive, and disseminated.

The No. 3 Sn-Pb-Zn ore vein transitions from medium-high to medium-low temperature from southwest to northeast. Sn and Pb-Zn mineralization dominate the SW and NE sections, respectively. While the deep mineralization is excellent, the shallow mineralization is weak. The fault fracture zone controls the ore vein, which extends in a wavy pattern along the strike and trend, with local expansion and contraction. The wall rock alteration is almost the same as that of No. 4. Ore minerals include galena, sphalerite, pyrite, and chalcopyrite, followed by cassiterite, wolframite, and arsenopyrite. Ore textures and structures were dominated by massive, disseminated, and veined, followed by allotriomorphic granular and metasomatic-relict (see Table 1 for mineral composition).

Based on fieldwork and petrographic observations, Sn-W polymetallic mineralization at Hongqiling was identified in three stages (S1–S3) (Figures 3–5):

S1_ (-W-Sn mineralization): the mineral assemblage comprises cassiterite, wolframite, scheelite, arsenopyrite, molybdenite, pyrite, chalcopyrite, and quartz;

S2_ (-Pb-Zn mineralization): mineral assemblage comprises chalcopyrite, pyrrhotite, galena, sphalerite, pyrite, quartz, and fluorite;

S3_ (-quartz-fluorite): mineral assemblage comprises quartz, fluorite, calcite, galena, sphalerite, and pyrite.

**Table 1.** Ore vein type and mineral assemblage of the Hongqiling W-Sn-Pb-Zn deposit.

| | Ore Vein | Type | Gangue Mineral | Ore Mineral | Texture of Ores | Ore Structure |
|---|---|---|---|---|---|---|
| W-Sn mineralization stage | 4 | Cassiterite Sulfide type | A small amount of quartz | Cassiterite dominates, with small amounts of wolframite, pyrrhotite, and chalcopyrite | Euhedral-hypidiomorphic granular, cataclastic | The massive and disseminated |
| | 101, 102, 103 | Cassiterite Quartz vein type | Quartz, fluorite, a small amount of calcite | Mainly wolframite, cassiterite, and arsenopyrite, followed by scheelite, molybdenite, chalcopyrite | Common crystallization, metasomatic-relict, euhedral-hypidiomorphic granular | Banded, massive, veined |
| Pb-Zn mineralization stage | 27 | Lead-Zinc Ore vein | Quartz, fluorite, and calcite | Chalcopyrite, and sphalerite dominate, followed by galena and pyrite | metasomatic-relict, allotriomorphic granular, idiomorphic granular | The massive and disseminated |
| | 3 | The mixed tin-lead-zinc ore vein | Quartz, fluorite, and calcite | Galena, sphalerite, pyrite, and chalcopyrite mainly, followed by cassiterite, wolframite arsenopyrite | allotriomorphic granular and metasomatic-relict | The massive, disseminated and veined |

| Mineralization stage / Mineral species | High temperature W-Sn mineralization (S1) | High-medium temperature Pb-Zn mineralization (S2) | Medium-low temperature Late mineralization (S3) |
|---|---|---|---|
| | W-Sn quartz veins | Sulfide veins | Quartz-fluorite-calcite veins |
| Quartz | ████████████ | ████████████ | ████████████ |
| Wolframite | ███████ | | |
| Scheelite | ████ | | |
| Cassiterite | ██████ | | |
| Molybdenite | ███ | | |
| Arsenopyrite | ████ ████ | | |
| Chalcopyrite | ─── ─── | ─────── | |
| Pyrrhotite | ██████ | █ | |
| Galena | | ────████─── | ─── ─── |
| Sphalerite | | ████───── | ─── ─── |
| Pyrite | ─── | ──────── | |
| Fluorite | ─── | ─── ─── | ─── ─── |
| Calcite | | | ── ── ── |

**Figure 3.** The mineral paragenetic sequence of the Hongqiling Sn-W polymetallic deposit.

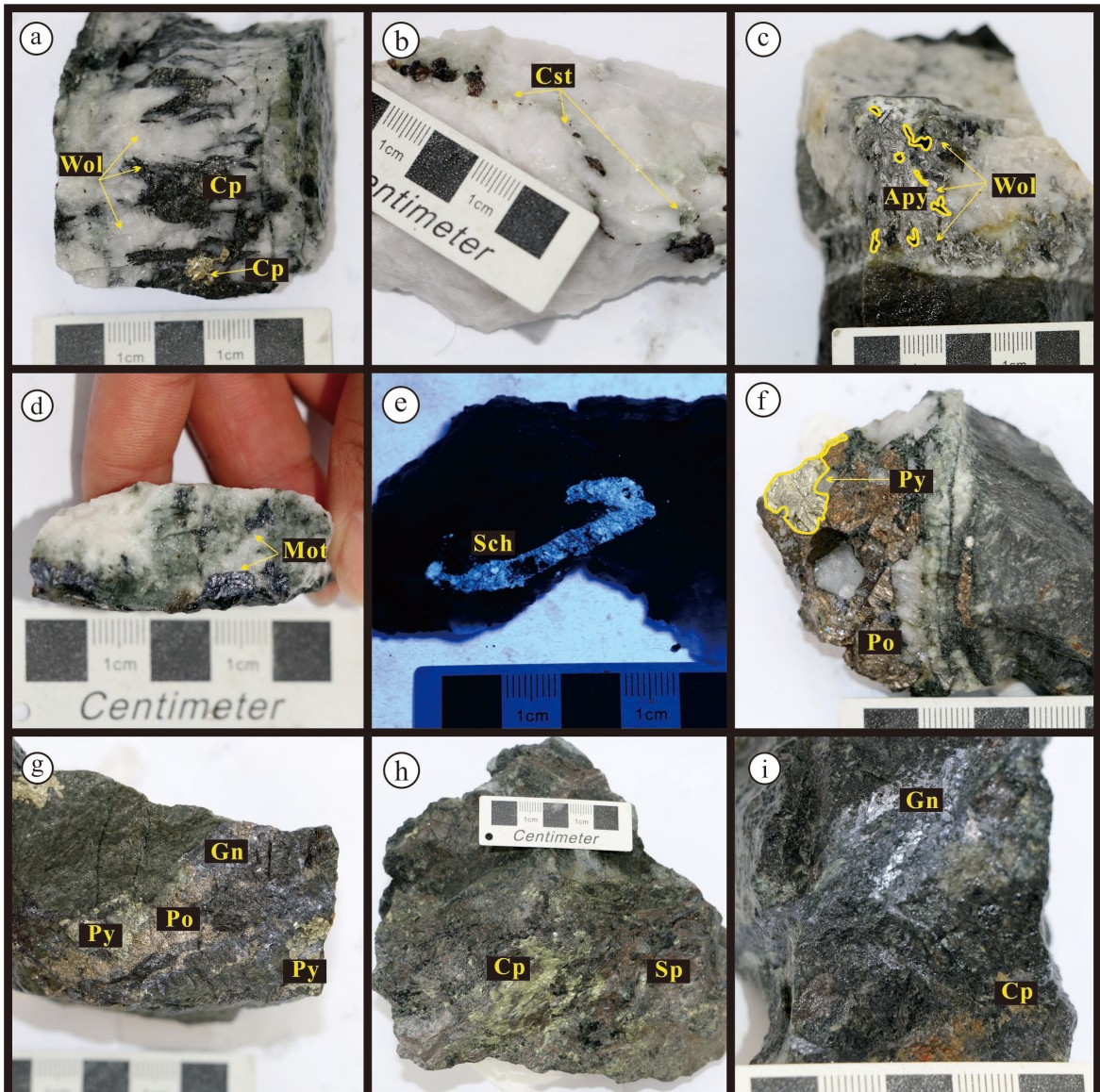

**Figure 4.** Photographs of representative Hongqiling Sn-W ore samples. (**a**) Euhedral wolframite intergrown with chalcopyrite in S1 wolframite-quartz veins; (**b**) cassiterite particles filled in S1 quartz veins; (**c**) wolframite intergrown with arsenopyrite in S1 wolframite-bearing quartz veins; (**d**) layered molybdenite in early high-temperature quartz veins; (**e**) fluorescence effect of scheelite; (**f**) tarnish pyrrhotite intergrown with pyrite in sulfide veins (S2 stage); (**g**) ore minerals in sulfide veins (S2 stage); (**h**) disseminated chalcopyrite intergrown with disseminated sphalerite (S2 stage); (**i**) disseminated galena intergrown with minor chalcopyrite (S2 stage). Wol: wolframite; Cp: chalcopyrite; Cst: cassiterite; Apy: arsenopyrite; Mot: molybdenite; Sch: scheelite; Py: pyrite; Po: pyrrhotite; Gn: galena; Sp: sphalerite.

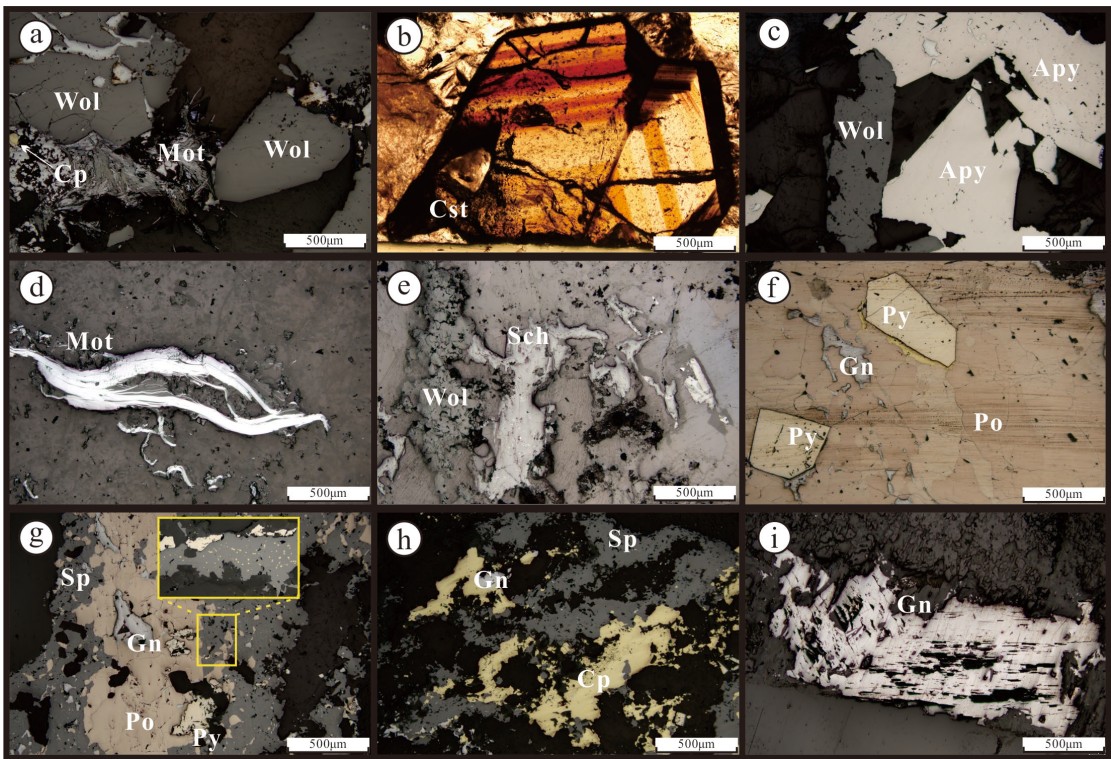

**Figure 5.** Photomicrographs of representative hand samples from the Hongqiling Sn-W polymetallic deposit. (**a**) Columnar wolframite with molybdenite, and chalcopyrite; (**b**) well-zoned cassiterite; (**c**) wolframite intergrown with arsenopyrite; (**d**) filamentous molybdenite in quartz vein; (**e**) wolframite intergrown with scheelite; (**f**) euhedral pyrite enclosed in pyrrhotite with minor galena; (**g**) pyrrhotite intergrown with pyrite, galena, and sphalerite with chalcopyrite disease; (**h**) chalcopyrite intergrown with sphalerite, locally visible galena; (**i**) galena deformed by external stress. Wol: wolframite; Cp: chalcopyrite; Cst: cassiterite; Apy: arsenopyrite; Mot: molybdenite; Sch: scheelite; Py: pyrite; Po: pyrrhotite; Gn: galena; Sp: sphalerite.

## 3. Sampling and Methodology

### 3.1. Sampling

The fluid inclusion samples were obtained from ore-related quartz, fluorite, calcite, and wolframite. Forty-five inclusions samples were ground from different mine levels of W-Sn mineralization, Pb-Zn mineralization, and late mineralization for the fluid inclusion study. The samples were mainly collected from veins at level 836 m (Vein No. 3, 101, 102, 103, and 27), 766 m (Vein No. 101, 102, and 103), 610 m (Vein No. 101, 102, and 103), and 420 m (Vein No. 4, 27, 101, 102, and 103). In this study, quartz, fluorite, and wolframite were analyzed to determine the mineralization temperature, and petrographic work was carried out based on fieldwork. The samples were cleaned, observed, and described, and inclusions sections were ground. The study area's essential characteristics of inclusions such as type, size, shape, and vapor–liquid ratio were confirmed under the microscope and photographed and recorded. Based on detailed microscopic observation, the inclusions sections were classified and soaked in acetone and alcohol to remove the gum on the back of the inclusions sections and cleaned. The representative inclusions should be circled for the area where they are located to facilitate the inclusion of microthermometric work and various analysis works.

### 3.2. Analytical Methods

#### 3.2.1. Fluid Inclusion Microthermometry

The analysis was performed at the Fluid Inclusion Laboratory of the Southwest Geological Survey, Kunming University of Science and Technology. The microthermometric

data of fluid inclusions in transparent minerals were obtained with a Linkam THMSG600 heating-freezing stage, with the analysis temperature range of $-196$ to $+600$ °C and accuracy of $\pm 0.1$ °C. Microthermometric data of the wolframite were obtained with a Linkam THMSG600 heating-freezing stage coupled to an Olympus BX51 infrared microscope in the inclusion room of the State Key Laboratory for Mineral Deposit Research, Nanjing University. The temperature range is the same, and the accuracy is $\pm 0.2$ °C below 30 °C, $\pm 0.5$ °C from 0 to 100 °C, and $\pm 2$ °C from 100 to 600 °C.

The ice-melting temperature was chosen first to avoid the bursting of fluid inclusions during the heating process, followed by the heating test homogenization temperature. During the analysis, the freezing and heating rate was controlled at 25 °C/min, and the rate was gradually dropped to 5 °C/min, 1 °C/min, and 0.2 °C/min when approaching the phase transition. The salinity of the vapor-liquid two-phase aqueous inclusions was obtained as the measured ice-melting temperature according to the freezing point–salinity relationship table of fluid inclusions freezing method [46]. In addition, the infrared thermometry of wolframite was limited by certain intrinsic factors. They limit the microscopic observation of inclusions in the sample, such as the significant increase in opacity of wolframite during the warming process. The dark areas in the line of sight are caused by the cleavage plane, and the low resolution is due to the long wavelength of infrared light. Therefore, the temperature rise and fall rates need to be controlled more strictly to obtain more accurate data.

### 3.2.2. Fluid Inclusion Laser Raman Spectroscopic Analysis

The analysis was performed at the Laser Raman Spectroscopy Laboratory of the Southwest Geological Survey, Kunming University of Science and Technology, with an RM$-$1000 laser Raman spectrometer. The analytical conditions include the light source of 514.5 μm, 10 s counting time, every 1 cm$^{-1}$ is counted once, 2 μm laser spot size, and 2 cm$^{-1}$ spectral resolution.

### 3.2.3. H–O Isotope Analyses

Representative samples of quartz (from S1 and S2) and wolframite (from S1) were selected for the analyses. Before the analyses, fresh quartz samples were hand-picked for crushing, sorting, and rinsing (with distilled water and anhydrous ethanol). Mineral grains free of impurities and coexisting minerals were selected with over 99% purity. The analyses were carried out at the Analytical Laboratory of Beijing Research Institute of Uranium Geology, China. Hydrogen isotopes were measured with a MAT253 Isotope Mass Spectrometer using the zinc reduction method, while oxygen isotopes were measured with a Delta V Advantage Vapor Isotope Mass Spectrometer using the BrF$_5$ method. The data were calibrated by V-SMOW, with the analytical accuracy of $\pm 0.2$‰ and $\pm 2$‰ for O and H isotopes, respectively.

### 3.2.4. LA-MC-ICP-MS Sulfur Isotope Analysis

The analysis was completed in the Guangzhou Institute of Geochemistry, Chinese Academy of Sciences, using a Thermo Scientific Neptune Plus Multi Collector (MS)-ICP-MS, coupled with a Resolution s155 193 nm laser ablation system. The analysis spots were selected from optical microscopic and SEM images. The analysis used 24 μm spot size, 4 J/cm$^2$ energy density, and 6 Hz frequency. He was the carrier gas to transport the aerosol into the MC-ICP-MS. The $^{32}$S and $^{34}$S signals were detected by the Faraday cup contemporaneously, with an integration time of 0.131 s. A total of 200 sets of data were collected, with a total time of ~27 s. Before the analysis, the instrument parameters were adjusted with sulfide standards HN, JX, and ZX to reach the optimal state. To minimize the matrix effect, sulfides similar to the sample matrix were used as the standard. The quality discrimination was corrected by the standard-sample-standard cross method.

## 4. Results

### 4.1. Fluid Inclusions

4.1.1. Petrography and Microthermometry of Fluid Inclusions

Based on optical and infrared observations, fluid inclusions are widespread in the quartz, wolframite, fluorite, and calcite from Hongqiling. According to the classification scheme of fluid inclusion phase changes at room temperature and during freezing and heating [47,48] and the fluid inclusion genetic classification [47,49], we classified our fluid inclusion samples into the following two types: (1) liquid-rich two-phase FIs (L-type) and (2) vapor-rich two-phase FIs (V-type). L-type FIs are dominant in quartz and occupy > 60% of the total inclusion numbers, while V-type FIs are subordinate and contribute ~ 40% of the total inclusion numbers. The inclusions in wolframite are mainly V-type, occasionally accompanied by pure vapor-phase. In this study, the microthermometric measurements were carried out either on the isolated inclusions thought to be primary [47] or on a cluster of FIs with similar heating-freezing behavior, which together represent fluid inclusion assemblages (FIA) [49]. In addition, three-phase inclusions are relatively rare in quartz, which have yet to be studied due to their small quantity.

4.1.2. Fluid Inclusions in Wolframite (S1)

For the convenience of the following statement, the vapor-rich two-phase inclusions widely distributed in wolframite are named W-type inclusions, and the two-phase inclusions in quartz that coexisted with them are named Q-type inclusions. Almost all the analyzed FIs are ellipsoid and 3–10 μm/20–55 μm in size and contain vapor bubbles of 50–80 vol%. Their distribution in wolframite is mostly isolated (Figure 6a,b,d,e) or as a cluster of inclusions along fissures and crystalline bands (Figure 6c,f). W-type FIs in wolframite of S1 homogenized to liquid, with Th of 258.6–316.9 °C and salinities of 3.8–8.9 wt.% NaCleqv.

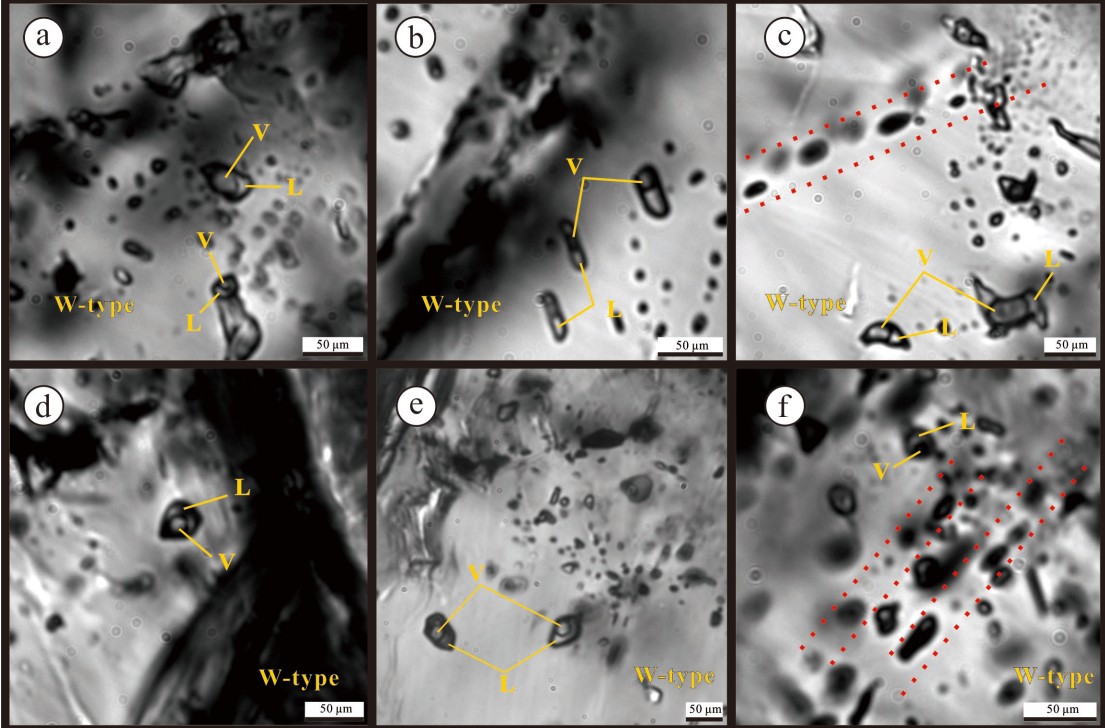

**Figure 6.** Infrared micrographs of fluid inclusions in wolframite from the Hongqiling Sn-W polymetallic deposit: (**a**,**e**) coexistence of W-type inclusions with different phase ratios; (**b**) combination of W-type inclusions distributed along a fissure; (**c**,**f**) combination of W-type inclusions distributed along the growth annulus and vapor-rich W-type inclusions; (**d**) isolated W-type inclusions.

### 4.1.3. Sn-W Quartz Vein Stage (S1)

The types of fluid inclusions in Sn-W quartz veins are mainly L-type inclusions. They are regular quadrilateral, triangular, irregular, or distributed in small clusters. Some isolated fluid inclusions or FIAs that are regarded as primary are selected to conduct microthermometry (Figure 7a,e,g). The long axis of the inclusions is 5–15 μm long, 10–35 μm, to 5–20 μm. The ice-melting temperature of L-type inclusions is −2.1 to −5.6 °C, with the corresponding salinity of 3.5–8.7 wt.% NaCleqv. The inclusions homogenized to the liquid phase at 236–358.8 °C. V-type inclusions are up to 70 μm long, with a vapor ratio of 30% to 70%. They have ice-melting temperatures of −3.5 to −7.2 °C, corresponding to a salinity of 5.7 to 10.7 wt.% NaCleqv, with an inclusion Th range of 284.5 to 377.6 °C, most of which are wholly homogenized to the liquid phase state and sporadically homogenized to the vapor phase (Figure 8).

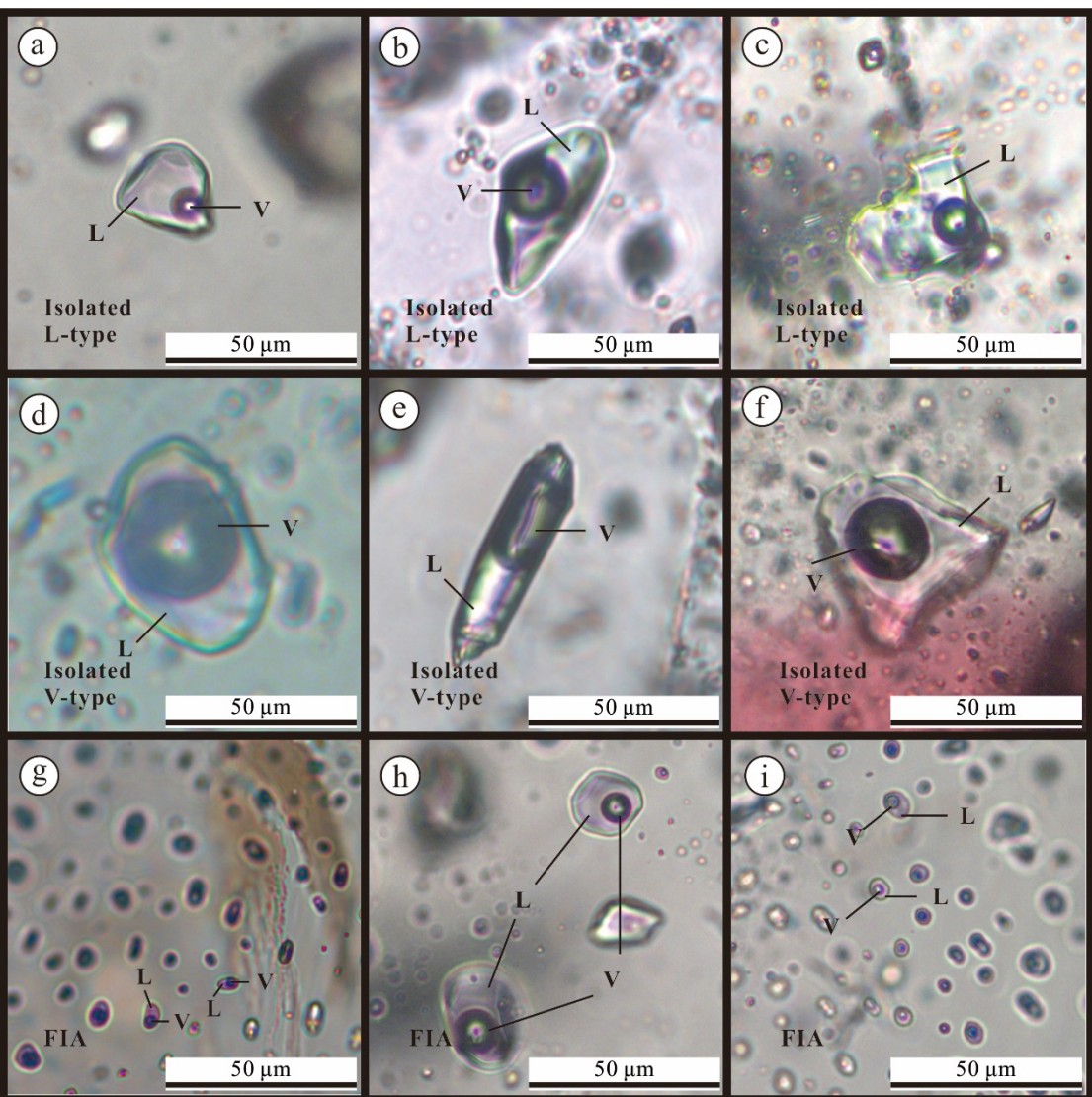

**Figure 7.** Microphotographs of fluid inclusions in quartz and fluorite from the Hongqiling Sn-W polymetallic deposit. (**a**–**c**) Isolated L-type inclusion; (**d**–**f**) isolated V-type inclusion; (**g**–**i**) a cluster of FIs with similar heating-freezing behavior, which together represent a fluid inclusion assemblage (FIA).

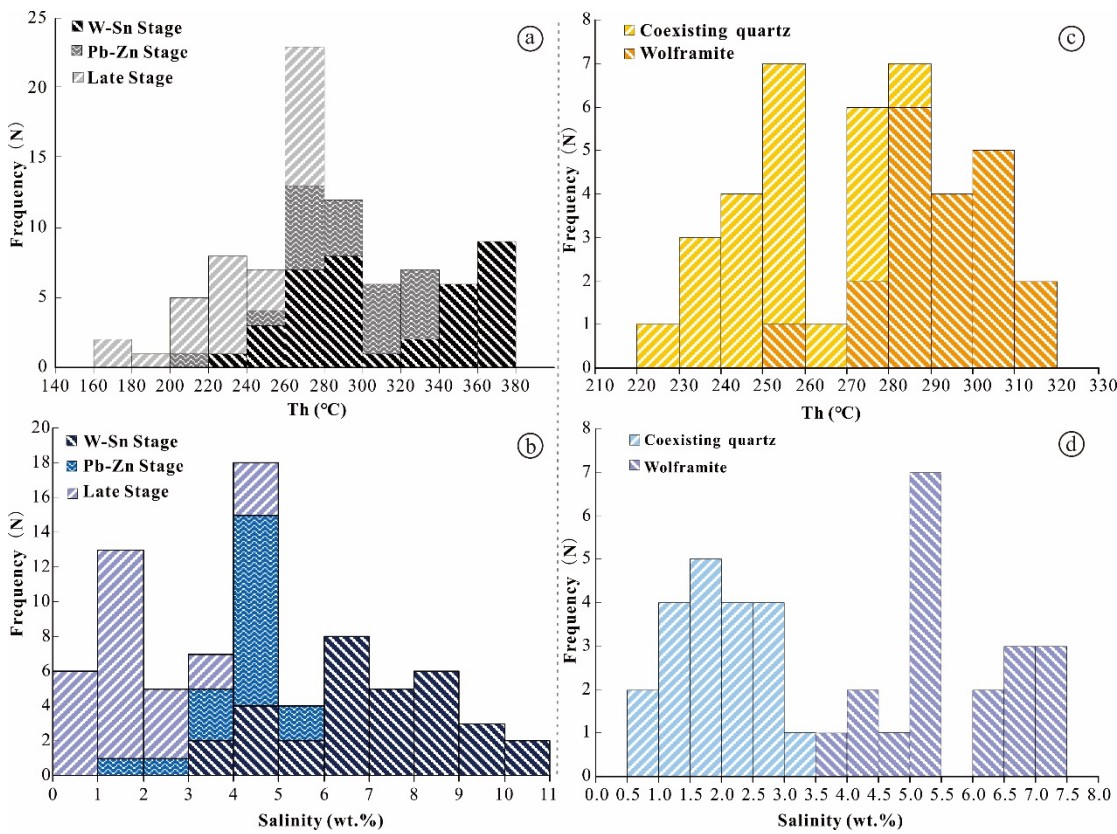

**Figure 8.** (**a**,**b**) Histogram of homogenization temperature and salinity at each mineralization stage. S1–S2 stage analysis minerals: quartz; S3 stage analysis minerals: quartz, fluorite, calcite; (**c**,**d**) histogram of homogenization temperature and salinity of wolframite and its coexisting quartz.

### 4.1.4. Pb-Zn Quartz Vein Stage (S2)

Inclusions in S2 quartz are mainly L-type. They are negatively crystalline, elongated, and distributed in small clusters that are about 3–15 μm and 20–25 μm long with vapor phase of 10%–40%. Some isolated fluid inclusions or FIAs that are regarded as primary are selected to conduct microthermometry (Figure 7b,d,i). The ice-melting temperature of L-type inclusions is −0.9 to −3.1 °C, corresponding to salinities of 1.6–5.1 wt.% NaCleqv. The inclusions homogenized to the liquid phase at 206.5–332 °C (Figure 8).

### 4.1.5. Quartz-Fluorite Vein Stage (S3)

Inclusions in fluorite are 15–70 μm long, with a vapor bubbles ratio = 18 to 80%. Additionally, some isolated fluid inclusions or FIAs considered primary are selected to conduct microthermometry (Figure 7c,f,h). They are overall larger and have more explicit phase boundaries than those in quartz, with ice-melting temperatures varying from −0.1 to −3.6 °C. The inclusions homogenized to the liquid phase at 170.9–328.7 °C, with corresponding salinity of 0.2–5.9 wt.% NaCleqv (Figure 8).

### 4.2. Laser Raman Spectroscopic Data

Based on petrographic observation, laser Raman spectra were performed on representative fluid inclusions' vapor-phase and liquid-phase compositions. The results show clear and broad $H_2O$ envelope peaks in both S1 and S2, where a strong $CO_2$ Fermi peak is accompanied by minor $N_2$ in the vapor phase of S1 inclusions (Figure 9a). The intensity of $CO_2$ spectral peaks in the liquid phase decreases. In contrast, the $N_2$ spectral peaks are still prominent (Figure 9b). The intensity of $CO_2$ and $N_2$ spectral peaks in both phases of the S2 inclusions continues to decrease compared with that of S1 (Figure 9c,d). Interestingly, despite the presence of $CO_2$ in FIs, no $CO_2$-bearing FIs were observed petrographically.

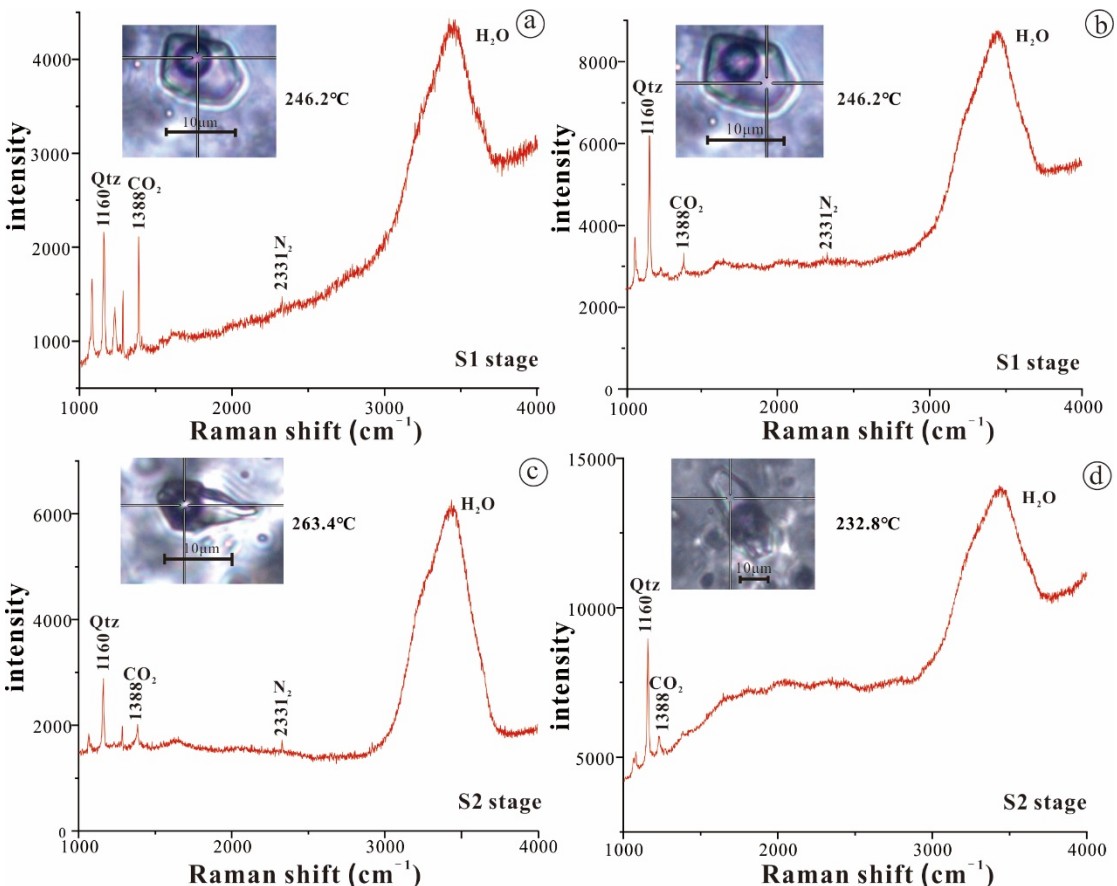

**Figure 9.** Laser Raman spectra of fluid inclusions in the Hongqiling Sn-W polymetallic deposit. (**a**): S1 stage fluid inclusion vapor phase; (**b**): S1 stage fluid inclusion liquid phase; (**c**): S2 stage fluid inclusion vapor phase; (**d**): S2 stage fluid inclusion liquid phase.

### 4.3. Fluid Density, Pressure, and Metallogenic Depth

#### 4.3.1. Fluid Density

The empirical formula for calculating the density of inclusions in $NaCl-H_2O$ fluid [50] can be expressed by:

$$\rho = a + b * T + c * T^2 \tag{1}$$

where $\rho$ = brine density (g/mL) and T = mean temperature (°C); a, b, and c are functions of salinity.

The calculated fluid density at Hongqiling is 0.68–0.87 g/mL (avg. 0.77 g/mL) for the W-Sn ore stage, 0.66–0.9 g/mL (avg. 0.78 g/mL) for the Pb-Zn ore stage, and 0.59–0.91 g/mL (avg. 0.82 g/mL) for the late ore stage. At the same time, combined with the homogenization temperature and salinity of the inclusion, the Th-S-D phase diagram of the $NaCl-H_2O$ system was drawn to support the calculated density value [51], which was consistent with the calculated value (Figure 10b). Overall, the ore fluids at Hongqiling are of low density, with density gradually increasing with decreasing temperature (Table 2; Figure 10a).

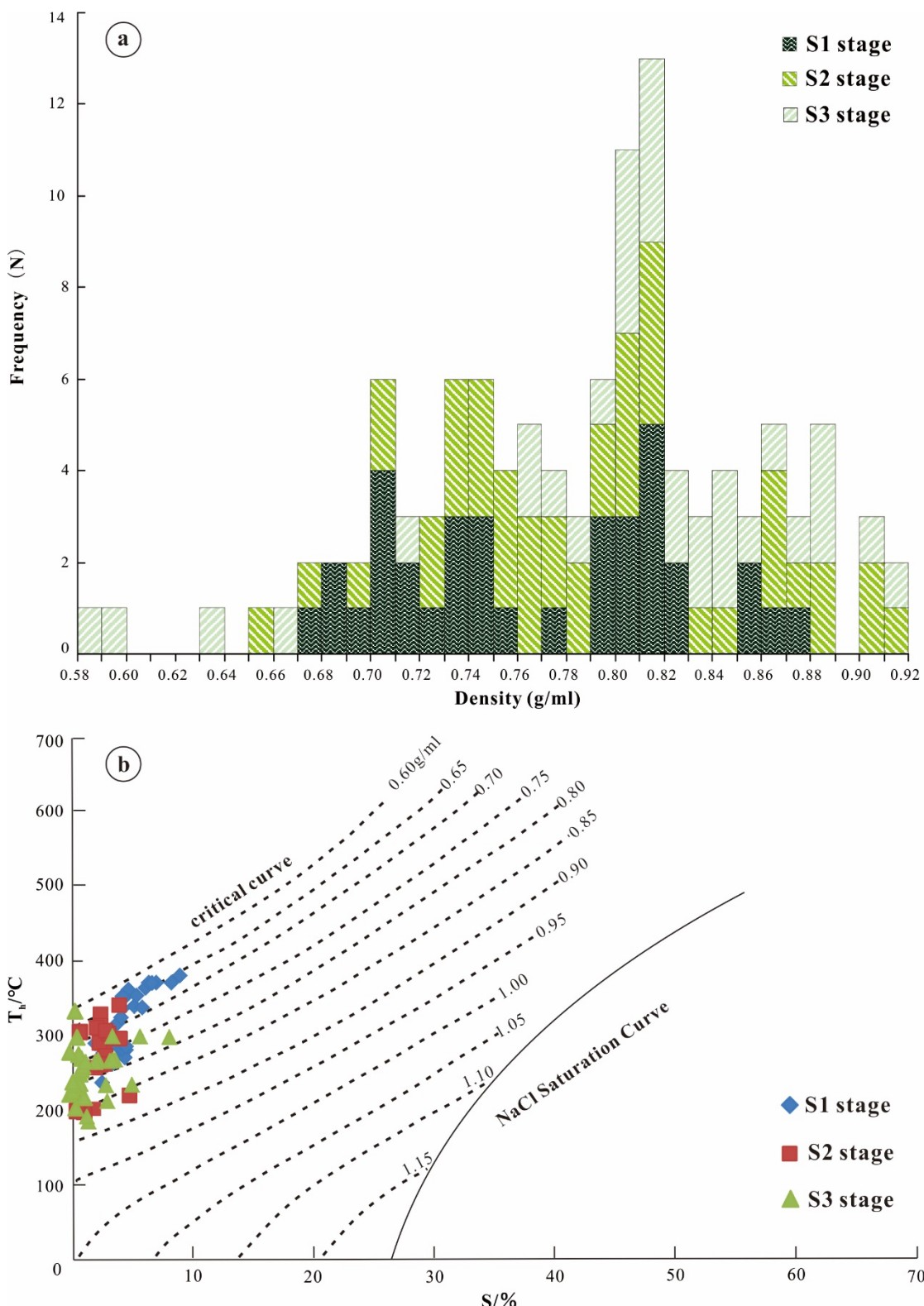

**Figure 10.** (**a**) Fluid density histogram of Hongqiling Sn-W polymetallic deposit; (**b**) phase diagram of Th-salinity-density in NaCl-H$_2$O system.

**Table 2.** Fluid inclusion data of the Hongqiling Sn-W polymetallic deposit.

| Mineralization Stage | Number of Tests | Host Mineral | Type | Th (°C) (avg) | Salinity (wt.% NaCleqv) (avg) | Density (g/mL) (avg) | Pressure (Mpa) (avg) | Depth (km) (avg) |
|---|---|---|---|---|---|---|---|---|
| W-Sn mineralization | 32 | Quartz | L + V | 236~377.6 (305.3) | 3.5~10.7 (8.07) | 0.68~0.87 (0.77) | 62.77~103.56 (82.39) | 2.09~3.44 (2.72) |
| Pb-Zn mineralization | 25 | Quartz | L + V | 206.5~332 (280.7) | 1.6~5.1 (3.84) | 0.66~0.9 (0.78) | 46.07~89.01 (69.78) | 1.54~2.97 (2.33) |
| Late stage of mineralization | 28 | Quartz and fluorite | L + V | 170.9~328.7 (246) | 0.2~5.9 (2.44) | 0.59~0.91 (0.82) | 23.97~81.4 (59.27) | 0.80~2.71 (1.98) |

4.3.2. Metallogenic Pressure and Depth Estimation

The formation pressure of gas-liquid two-phase inclusions was calculated using homogenization temperature according to the empirical formula [52]:

$$P = (219 + 2620 \times \omega) \times T \div (374 + 920 \times \omega) \tag{2}$$

where P = $10^5$ Pa, $\omega$ = fluid inclusion salinity (mass %), and $T$ = fluid inclusion homogenization temperature (°C). The calculated ore fluid pressure is 62.77–103.56 MPa (avg. 82.39 MPa) for the W-Sn ore stage, 46.07–89.01 MPa (avg. 69.78 MPa) for the Pb-Zn ore stage, and 23.97–81.4 MPa (avg. 59.27 MPa) for the late ore stage (Table 2).

The depth of mineralization can be estimated by:

$$H = P \times 10^{-5}/300 \tag{3}$$

where H = depth of mineralization (km) and P = mineralization pressure ($10^5$ Pa) [53]. As listed in Table 2, the mineralization depth at Hongqiling is 2.09–3.44 km (avg. 2.72 km) for the W-Sn ore stage, 1.54–2.97 km (avg. 2.33 km) for the Pb-Zn ore stage, and 0.80–2.71 km (avg. 1.98 km) for the late ore stage. This variation indicates that the Hongqiling W-Sn-Pb-Zn mineralization occurred at a shallow depth.

*4.4. H–O Isotope Compositions*

The ore-bearing quartz vein samples used for H-O isotope analysis represent the ore-stage fluid H-O isotope compositions. The quartz $\delta D_{smow}$ value varies from −51.5‰ to −76.6‰ (avg. −66‰), varying $\delta^{18}O_{v\text{-}smow}$ value (1.4–14.7‰): avg. 11.18‰ for the W-Sn ore stage, avg. 2.27‰ for the Pb-Zn ore stage (Table 3). According to the equation $1000\ln\alpha_{quartz\text{-}water} = 3.38 \times 10^6\ T^{-2}$–3.4 [54] and the wolframite-water balance fractionation equation of [55], the calculated fluid $\delta^{18}O$ value (from quartz) = −0.9 to −12.9‰ (avg.−3.2‰) and the average values are −3.2‰ and 11.9‰ in the different ore stages. The calculated fluid $\delta^{18}O$ value (from wolframite) is 5.1–6.5‰. In the $\delta^{18}O_{water}$−$\delta D_{smow}$ plot (Figure 11), the S1 stage shows a mixture of magmatic and meteoric water. All the $\delta D_{smow}$ values are within the magmatic water range, but the fluid $\delta^{18}O$ values are lower than the magma water. This shows meteoric water incursion that changed the fluid $\delta^{18}O_{v\text{-}smow}$ value (less so for the fluid $\delta D_{smow}$ value). In contrast, the $\delta D_{smow}$ value (S2 stage) is still within the magmatic water range, and the $\delta^{18}O$ water value deviates more from the original magma.

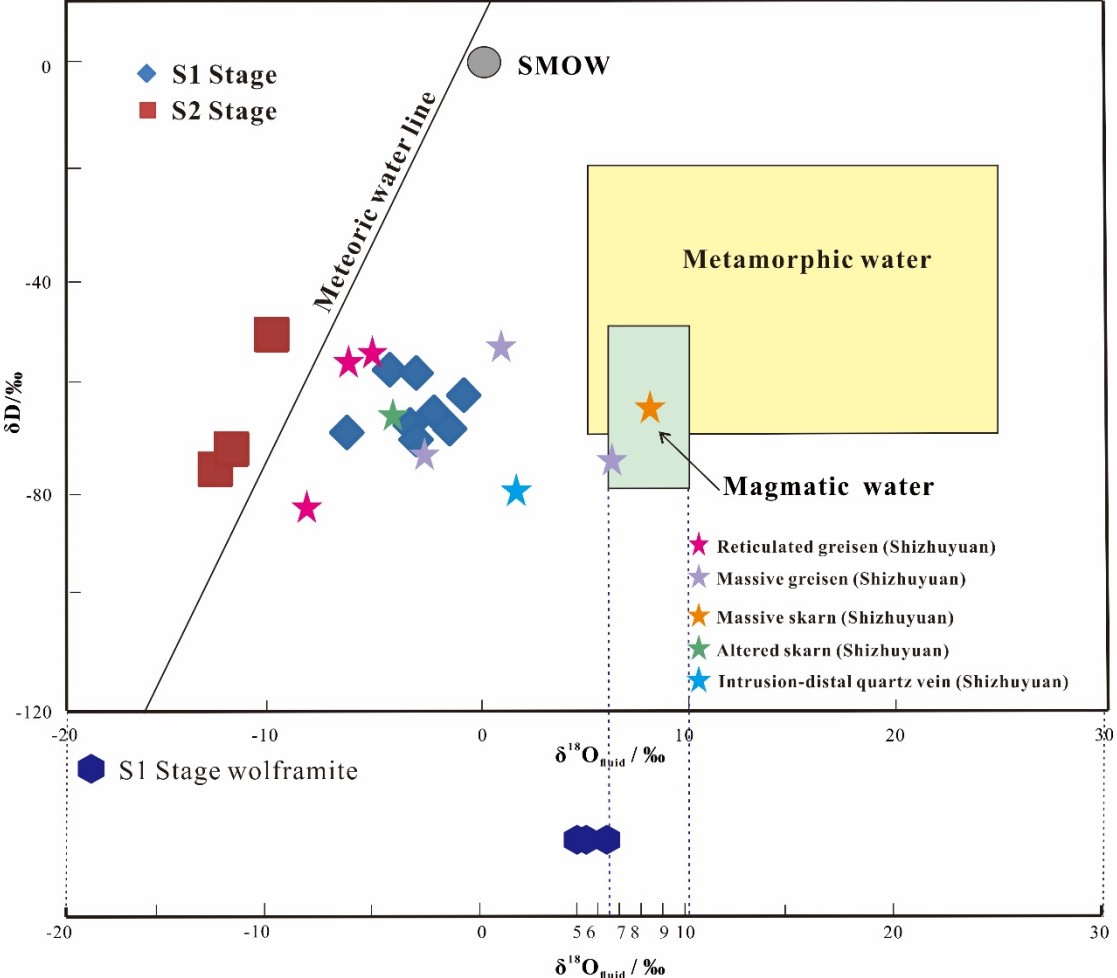

**Figure 11.** The plot of $\delta^{18}O$-$\delta D$ shows the calculated fluid composition of quartz veins in different stages. The primary magmatic water field, metamorphic water field, and meteoric water line are from [56]. SMOW = Standard Mean Ocean Water.

**Table 3.** Oxygen isotope data and the calculated fluid isotope compositions of the Hongqiling Sn-W polymetallic deposit.

| Sample Number | Ore Vein | Mineral | Th (°C) | $\delta D_{V\text{-}SMOW}$ (‰) | $\delta^{18}O_{V\text{-}SMOW}$ (‰) | $\delta^{18}O_{fluid}$ (‰) |
|---|---|---|---|---|---|---|
| FLR8-3 | | Quartz | 257 | −58.2 | 10.9 | −4.5 |
| HQR14 | | Quartz | 321 | −66 | 10.6 | −2.4 |
| HQR15-8 | | Quartz | 266 | −68.3 | 11.5 | −3.5 |
| HQR16 | | Quartz | 329 | −69.1 | 11.1 | −1.6 |
| HQR17 | W-Sn | Quartz | 278 | −58.7 | 11.3 | −3.2 |
| FLR23-2 | | Quartz | 253 | −62.8 | 14.7 | −0.9 |
| HQR15-7 | | Quartz | 278 | −71.1 | 11.3 | −3.2 |
| HQR15-1 | | Quartz | 285 | −69.8 | 7.7 | −6.6 |
| HQR2-1 | | Quartz | 278 | −76.6 | 1.6 | −12.9 |
| HQR2-2 | Pb-Zn | Quartz | 291 | −51.5 | 3.8 | −10.2 |
| FLR26 | | Quartz | 286 | −73.9 | 1.4 | −12.8 |
| HQR14 | | Wolframite | 321 | — | 4.1 | 5.7 |
| HQR16 | W-Sn | Wolframite | 329 | — | 4.9 | 6.5 |
| HQR17 | | Wolframite | 278 | — | 3.5 | 5.1 |

**Table 3.** *Cont.*

| | | | | Reference Data (Shizhiyuan) | | |
|---|---|---|---|---|---|---|
| Data Source | Rock Type | Mineral | Th (°C) | $\delta D_{V\text{-SMOW}}$ (‰) | $\delta^{18}O_{V\text{-SMOW}}$ (‰) | $\delta^{18}O_{fluid}$ (‰) |
| Wu et al., 2016 [57] | Greisen vein | Quartz | 225 | −83 | 2.5 | −8.2 |
| | | Fluorite | — | −55 | — | −5.8 |
| Lu, 2003 [58] | | Fluorite | — | −56 | — | −6.3 |
| Wu et al., 2016 [58] | Massive greisen | Quartz | 360 | −77 | 10.6 | 5.1 |
| | | Fluorite | — | −73 | — | −2.9 |
| Lu, 2003 [58] | | Fluorite | — | −52 | — | 1.2 |
| Lu, 2003 [58] | Massive skarn | Quartz | — | −64 | — | 8.5 |
| Wu et al., 2016 [57] | Altered skarn | Quartz | 185 | −65 | 8.3 | −4.2 |
| | Quartz veins located in the distant | Quartz | 225 | −83 | 13.6 | 2.8 |

*4.5. In Situ Sulfur Isotope Composition*

In this study, chalcopyrite and sphalerite from three samples were analyzed, and the results are in Table 4. The $\delta^{34}S$ values are −5.91 to 1.84‰ (avg. −0.93‰) for chalcopyrite and −2.58 to 0.83‰ (avg. −0.37‰) for sphalerite.

The petrographic study shows that the metal mineral assemblage in the study area is simple, and the sulfur-bearing mineral is only sulfide with no sulfate. This indicates that the sulfur isotope composition of the metal sulfides can approximate the total sulfur isotope composition in the ore-forming fluid [59,60]. The $\delta^{34}S$ value of the Hongqiling Sn-W polymetallic ore is narrow-range and mimics the mantle sulfur value (Figure 12), which implies a probable magmatic sulfur source (e.g., from the Qianlishan granite).

**Table 4.** In situ S isotope data of the Hongqiling Sn-W polymetallic deposit.

| Sample Number | Text Point Number | Mineral | $^{34}S$ (‰) | Sample Number | Mineral | $^{34}S$ (‰) | Reference Data |
|---|---|---|---|---|---|---|---|
| FLR23-2 | CP-1 | Chalcopyrite | −5.91 | 558-43 | Bismuthinite | 2.2 | |
| | CP-2 | Chalcopyrite | −1.31 | 490-70 | Bismuthinite | 3.2 | |
| | CP-3 | Chalcopyrite | −2.45 | 514-6 | Bismuthinite | 3.1 | |
| | CP-4 | Chalcopyrite | −2.19 | 490-120 | Bismuthinite | 1.4 | |
| FLR26 | SP-1 | Sphalerite | −2.58 | 490-121 | Bismuthinite | 1.8 | [61] |
| | SP-2 | Sphalerite | 0.83 | 490-131 | Bismuthinite | 1.5 | |
| | SP-3 | Sphalerite | 0.64 | 490-34 | Pyrite | 7.0 | |
| | CP-1 | Chalcopyrite | 1.80 | 490-38 | Pyrite | 6.2 | |
| | CP-2 | Chalcopyrite | 1.71 | Shi-2 | Pyrite | 4.9 | |
| | CP-3 | Chalcopyrite | 1.84 | — | Pyrite | 5.1 | |
| | | | | — | Pyrite | 4.8 | |
| | | | | D96 | Pyrite | 6.8 | |
| This study | | | | C650 | Galena | −5.2 | [62] |
| | | | | C660 | Galena | −4.9 | |
| | | | | E 7 | Galena | −7.2 | [40] |
| | | | | Shi 1 | Galena | −4.4 | |
| | | | | Shi 6 | Galena | −3.1 | |

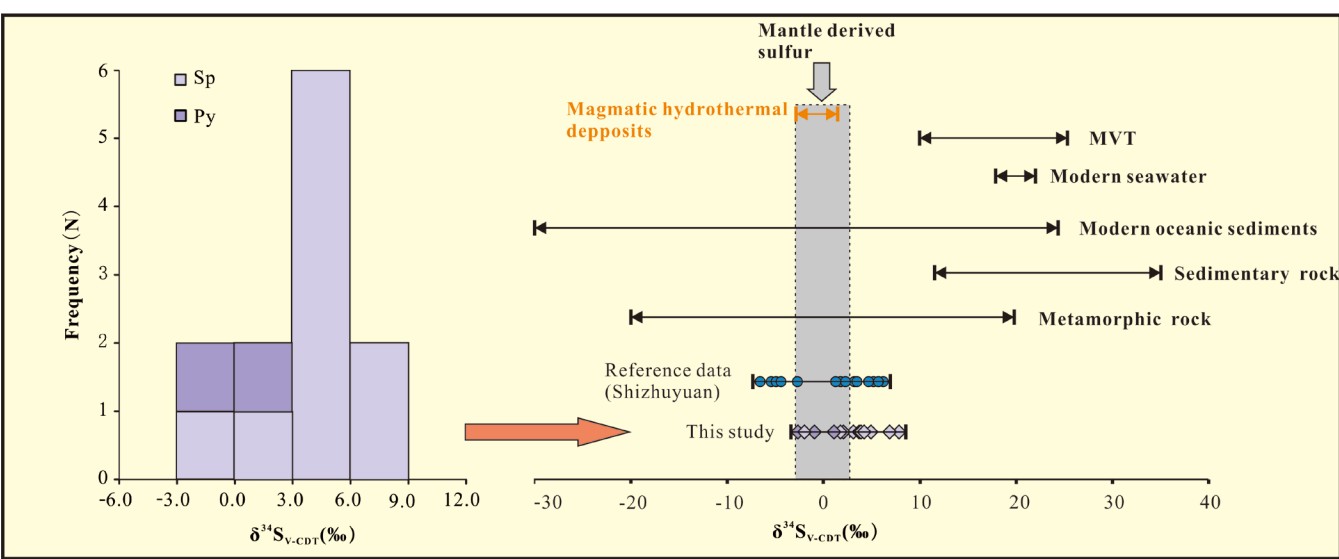

**Figure 12.** Comparison of sulfur isotope compositions of Hongqiling Sn-W polymetallic deposit and Shizhuyuan W-Sn polymetallic deposit with mantle-derived sulfur in the same period (isotopic data range from [63]).

## 5. Discussion

### 5.1. Ore-Forming Fluid Source and Evolution

#### 5.1.1. Ore-Forming Fluid Source

The natural sulfur isotope undergoes a variety of changes, with three major reservoirs being the most prevalent [64]. One is mantle-derived sulfur, whose $^{34}$S value is near 0, within ±3‰, showing an apparent tower distribution; the second is seawater sulfur, which has a relatively stable characteristic (+20‰) with large changes in sulfur isotope values; the third is biological sulfur, which is mainly characterized by a considerable negative value of $^{34}$S. As listed in Table 4 and Figure 12, the $\delta^{34}$S values of sulfides (chalcopyrite and sphalerite) are between–5.91 and 1.84‰. Except for one analysis point (FLR23-2 CP-1) that is characterized by a meager $\delta^{34}$S value ($\delta^{34}$S = −5.91‰), all the $\delta^{34}$S values of sulfides from the Hongqiling deposit are concentrated within the range of −2.58 to 1.84‰, which is consistent with the magmatic-hydrothermal deposits (−3 to 1‰ [63]), indicating that the sulfur in the Hongqiling Sn-W polymetallic deposit is probably from the Qianlishan granitic magma.

The calculated fluid $\delta^{18}$O (−12.8 to −0.9‰) is substantially lower than the $\delta^{18}$O value of magmatic water from the Qianlishan granite (8.5–12.5‰ [17]). It suggests that the ore fluids originated from magmatic water and were mixed with other fluids [65], which are characterized by $\delta^{18}$O < 0‰, consistent with previous research (Table 3; Figure 11). The H-O isotope data points of the main-ore S1 stage fall between the primary magmatic water and meteoric line, with a trend toward the latter. The fluid $\delta^{18}$O value of the S2 stage continues to decrease, but its δD value is comparable to that of S1. This implies that the meteoric water incursion is becoming more intense and that there was no new magmatic fluid injection.

In comparison to other Sn polymetallic deposits in the Nanling region (e.g., the Shizhuyuan ultra-large W-Sn polymetallic deposit), the Shizhuyuan deposit is made up of an intrusion-proximal skarn-greisen W-Sn-Mo-Bi deposit and intrusion-distal Pb-Zn-Ag vein-type deposit (Figure 11). Its reticulated greisen, retrograde-altered skarn, massive greisen, and quartz veins located in the distant contact zone is in the same area as the H-O isotopic projection point of the Hongqiling mine. The fluid $\delta^{18}$O and δD values are comparable to those found in this region. In comparison, the massive skarn data points all fall into the magma water field, implying that no meteoric water mixing had occurred. Given the long range from the ore vein, the reticulated greisen, the altered skarn on the outside,

and the quartz veins in the far contact zone extending to the northeast of the study area (the Congshuban area) have all undergone some degree of fluid mixing. Simultaneously, such features have been reported from the Jinchuantang Sn-polymetallic deposit, which is spatially adjacent to the Shizhuyuan deposit and has similar mineralization characteristics [58]. As a result, meteoric water plays a larger role in mineralization and is more significant at Hongqiling. It enters the fluid evolution process at the early stage of the mineralization process, indicating that the Hongqiling deposit is further away from the Qianlishan granite than the Shizhuyuan and Jinchuantang deposits.

The proportion of meteoric water in the fluid mixing regime can be estimated from the oxygen isotope plot [66]. As illustrated in Figure 13, the coupling trends of $\delta^{18}O$ value and temperature in quartz and wolframite differ depending on the mineralization mechanism: when the original ore fluid is mixed with a low-$\delta^{18}O_{fluid}$, the $\delta^{18}O$ value of the resultant fluid decreases; when the fluid boils, the $\delta^{18}O$ value increases with fluid evolution [67]. As described in Section 5.2.1, there is no evidence of fluid boiling in the primary ore stage, and thus, fluid boiling may have played only a minor role in the ore deposition. Our oxygen isotope data from Hongqiling show that the $\delta^{18}O$ value in both quartz and wolframite falls into the mixing evolution curve. In terms of homogenization temperature and $\delta^{18}O$ value, the rectangular box represents the area where the fluid inclusions in wolframite and quartz are coupled. In the early ore stage, the $X_A$ value fluctuates between 0.1 and 0.2 indicating only 10%–20% meteoric water incursion, as shown in Figure 13. The maximum $X_A$ value in the Pb-Zn ore stage is 0.8, indicating significant meteoric water mixing, which is consistent with the conclusion from the fluid analysis results.

### 5.1.2. Ore-Forming Fluid Evolution

Following the microthermometric results, the temperature, salinity, and vapor phase component steadily decrease with the mineralization from the W-Sn ore stage (236–377.6 °C) to the Pb-Zn ore stage (206.5–332 °C) to the late ore stage (170.9–328.7 °C) (Figure 8). Additionally, we hypothesized a broad dispersion of mean temperature for the inclusions in quartz by evaluating the scatter distribution of the mean temperature and salinity in the three phases, which suggests that the fluid activity connected to quartz was multiphase and complicated.

Heinrich (1990) suggested that fluid mixing can best explain the W-Sn ore deposition, with water–rock interaction or pressure reduction having a minor effect [68]. According to this study's Th-salinity scatter plot (Figure 14), fluid mixing is likely what caused the deposition of W-Sn ore because the mean temperature of S1 fluid inclusions is positively associated with salinity. This phenomenon is in line with the widespread reports of several Sn-W deposits around the world [69]. The W-Sn ore formation occurred in a more stable, higher pressure, and higher temperature environment than S2 and S3, as evidenced by the higher fluid temperature and salinity, lower fluid density, and greater fluid pressure in the W-Sn ore fluid when compared to S2 and S3 [70]. As previously stated, fluid mixing occurred shortly after the onset of quartz vein emplacement, resulting in rapid cooling and the consequent wolframite precipitation. Quartz, on the other hand, is a common mineral at Hongqiling that forms from S1 to S3. The fluid temperature range for S2 is broad, but the salinity range is narrow. This suggests that simple fluid cooling may have been crucial to the Pb-Zn sulfide ore formation. Xie et al. (2020) demonstrated that a significant decrease in mineralization temperature in the local range is required for metal sulfide deposition [71]. Reed (2006) used numerical simulation to show that Pb-Zn-Fe would precipitate when the metal chloride complexes were destabilized by temperature changes [72].

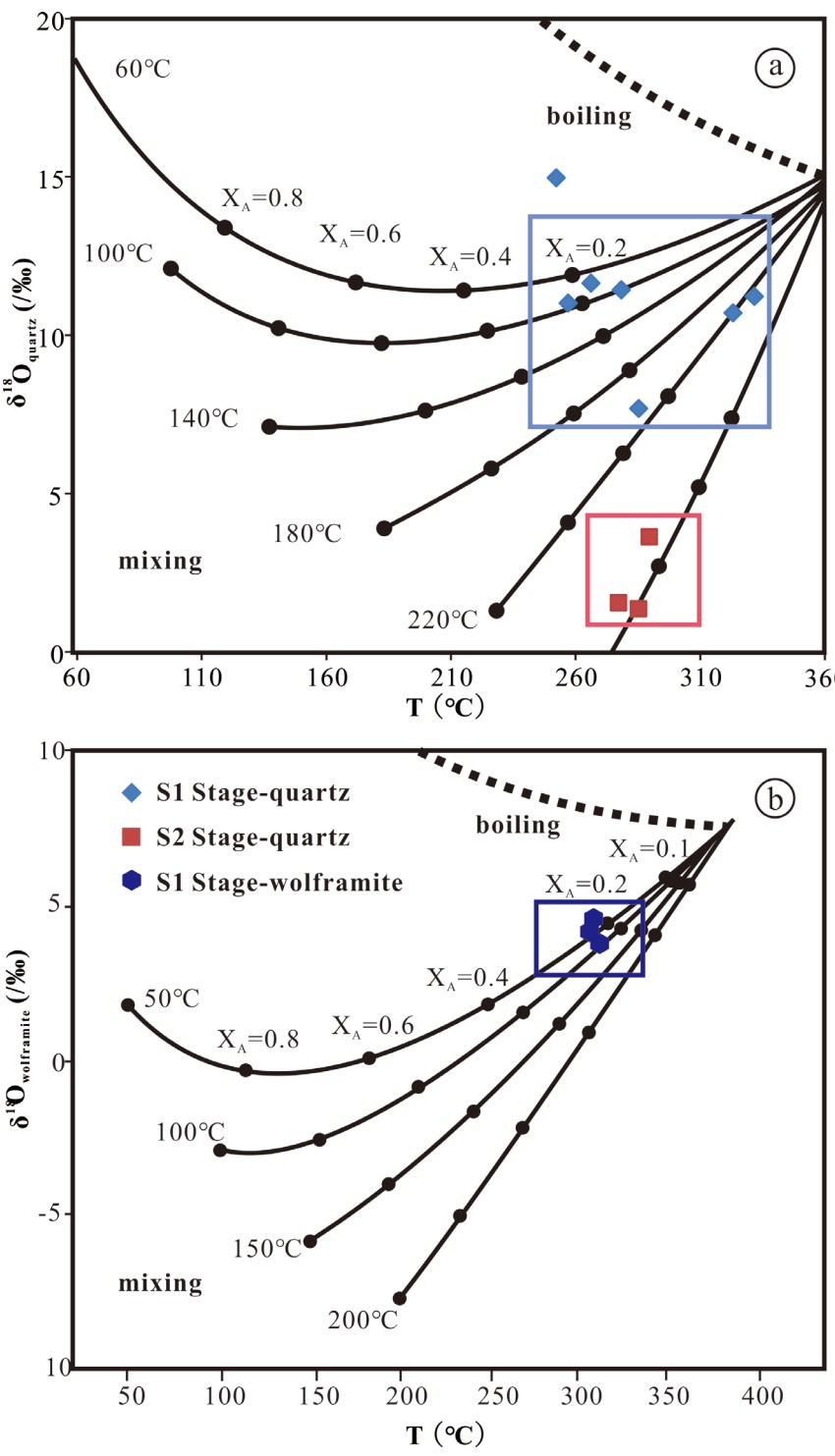

**Figure 13.** Effect of boiling and mixing of quartz (**a**) and precipitated wolframite (**b**) and on oxygen isotope composition at different temperatures (number $X_A$ is the mass fraction of atmospheric precipitation). Oxygen isotope trends of wolframite and quartz from [66].

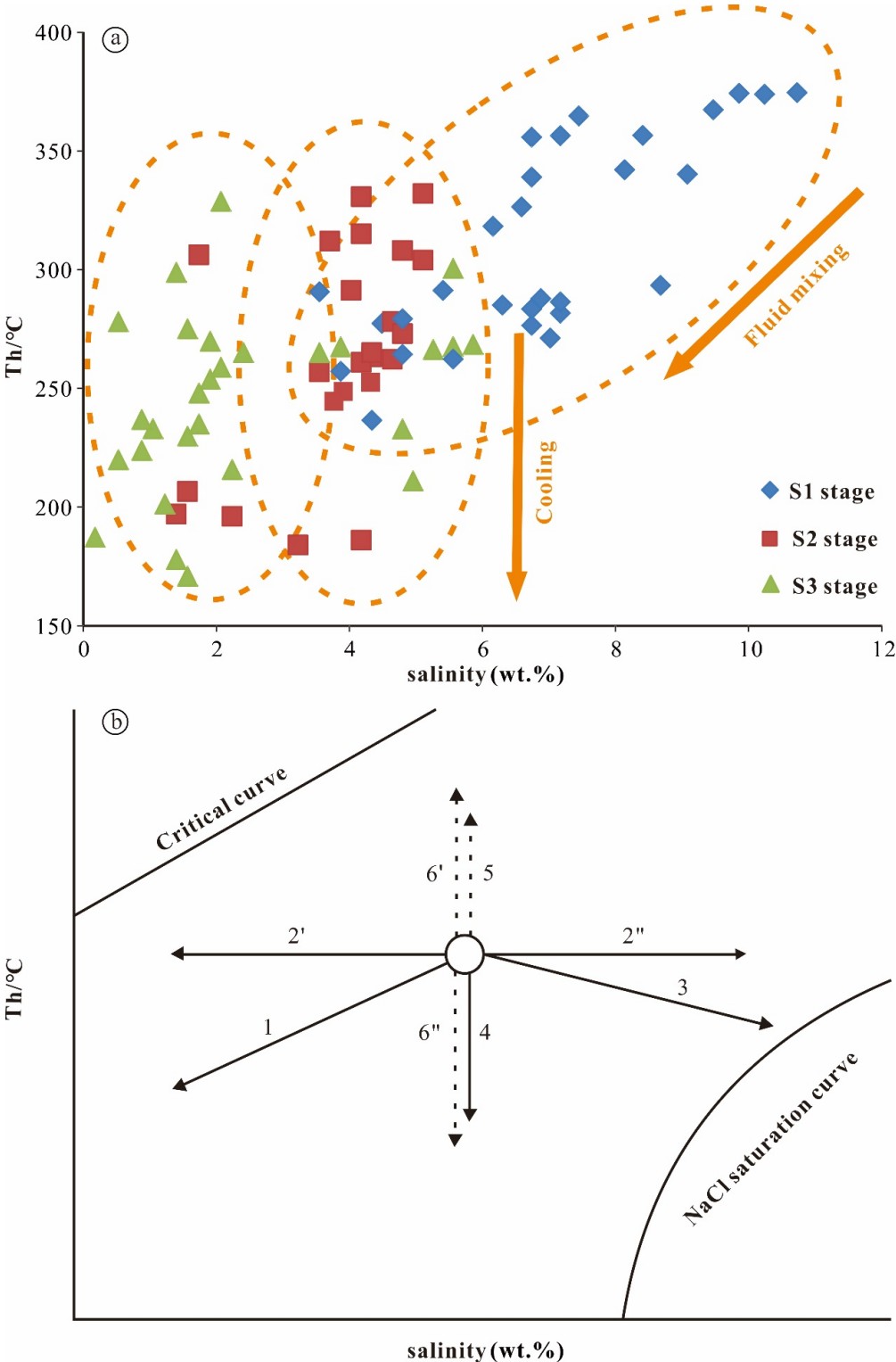

**Figure 14.** (**a**) Homogenization temperature vs. salinity plots of fluid inclusions in quartz from the Hongqiling deposit; (**b**) various fluid evolution trends in the Th-salinity bivariate plot. Trend 1 indicates that the original fluid is mixed with the cooler and lower salinity fluid; trends 2′ and 2″ represent the results of isothermal mixing of fluids with different salinity; trend 3 shows that the boiling of fluid leads to the increase in salinity of residual liquid phase; trend 4 represents pure cooling of fluid; trend 5 represents the leakage of inclusions during heating; trend 6 shows the necking phenomenon of inclusions [48].

At Hongqiling, faults and joints are well developed. W-Sn ore veins are controlled by the NE-trending faults (F4, F101, F102, and F103), and the major compression-strike-slip fault associated with the anticline at Hongqiling provides good open space for quartz-vein-type W-Sn-Pb-Zn mineralization [73]. As a result, we proposed the following scenario for Hongqiling's fluid evolution. During the ascent of metal-rich magmatic fluid, the mixing with 10%–20% meteoric water occurred at the early ore stage. When the temperature dropped to ~380 °C, quartz was first precipitated and fluid mixing continued, with the meteoric water percolated down along the faults/fractures and carrying a significant amount of metal elements in the wall rocks, which caused the fluid temperature to drop even more and prompted the precipitation of wolframite and other high-temperature minerals at ~310 °C; the W-Sn mineralization ended at ~260 °C. As the mineralization continued into the Pb-Zn sulfide ore stage, the hydrothermal-meteoric water mixing ratio increased up to 80%, which further cooled the fluid and reduced the ore metal solubility, causing large-scale intrusion and distal Pb-Zn-Ag ore precipitation at ~280 °C; the late-stage quartz veining may have occurred until ~170 °C. As a result, at Hongqiling, fluid mixing is most likely the primary ore deposition mechanism. The large W-Sn polymetallic deposits in the Nanling region were all thought to have been subjected to varying degrees of fluid mixing. However, Hongqiling differs from other deposits in Nanling due to the properties of its wall rocks (as described below). The clastic wall rocks produce fractures more readily, providing open space for fluid transport and mixing, and thus the degree of fluid mixing is greatest at Hongqiling.

### 5.2. Fluid Processes in Quartz and Coexisting Wolframite
5.2.1. Mechanism of Wolframite Mineralization

Opaque ore minerals were first studied by Campbell in 1984 using infrared microscopy, and since then, fluid inclusions in opaque minerals have been studied by several geologists [74,75]. These studies revealed a difference in the homogenization temperature between the ore minerals and the gangue minerals that coexisted, demonstrating that determining the temperature of fluid inclusions hosted in ore minerals can directly provide accurate data for estimating physicochemical conditions of metal precipitation [76]. In order to precisely ascertain the wolframite precipitation mechanism, infrared microthermometry of wolframite inclusions was performed in this investigation. All microthermometric results are provided in Table 5.

**Table 5.** Homogenization temperature and salinity data of wolframite and coexisting quartz.

| Host Minerals | Type of FIs | N | Th (°C) (avg) | Salinity (wt.% NaCleqv) (avg) |
|---|---|---|---|---|
| quartz | Iwq-type | 20 | 229.8~279.4 (253.8) | 0.7~4.3 (1.9) |
| wolframite | Iw-type | 20 | 258.6~316.9 (293.2) | 3.8~8.9 (5.8) |

Despite the fact that wolframite and quartz coexist spatially, the homogenization temperature and salinity of W-type inclusions are significantly higher than those of Q-type inclusions (Figure 8). This suggests that the coexisting wolframite and quartz precipitated separately at Hongqiling, with wolframite precipitating before quartz [76–78]. Field and petrographic observations imply that: (1) wolframite crystals generally occur on the vein margin, indicating that wolframite was formed before quartz (Figure 15a–c [76]); (2) the quartz has more inclusion types than wolframite. This tends to suggest that the quartz underwent multistage fluid evolution, resulting in numerous inclusions.

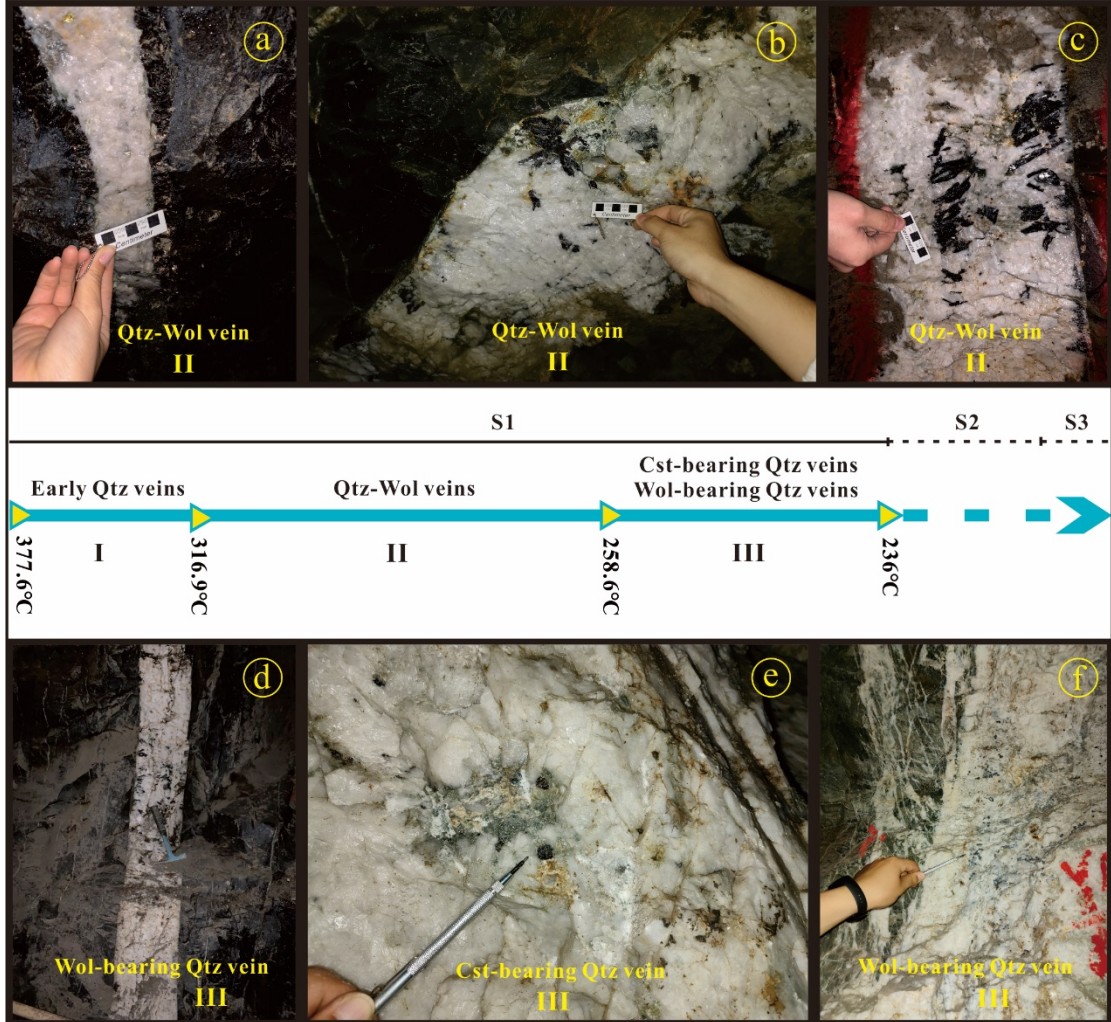

**Figure 15.** Deposition timeline of wolframite in the S1 mineralization stage. (**a**–**c**) Quartz-wolframite veining; (**d**,**f**) wolframite-bearing quartz veining; (**e**) cassiterite-bearing quartz veining.

In general, the wolframite precipitation mechanism includes fluid boiling or immiscibility, cooling, and mixing [79–81]. W is more soluble in acidic, saline, and reducing fluids, according to experimental studies, and it can be transported in fluids as simple tungstates [77]. As a result, fluid cooling or pH rise is essential for W precipitation. Higgins (1980) proposed that W can be transported in carbonate complexes, implying that an increase in pH caused by $CO_2$ loss could result in W precipitation [82]. However, Manning and Henderson (1984) showed that W precipitation is not directly related to degassing $CO_2$-rich fluids [83]. Therefore, an increase in PH is not the direct cause of W precipitation. Besides that, Drummond and Ohmoto (1985) emphasized the importance of fluid boiling in the precipitation of wolframite ore [84]. The relatively low fluid salinity of the S1 wolframite-quartz veining suggests that fluid mixing was more likely to occur than boiling. The homogenization temperature–salinity plot (Figure 16) clearly shows that the wolframite homogenization temperature is correlated positively with salinity and has a narrow salinity range, indicating that fluid mixing and cooling occur. However, there was no petrographic evidence of an immiscible inclusion group in our samples, which could be attributed to wolframite precipitation prior to the fluid boiling.

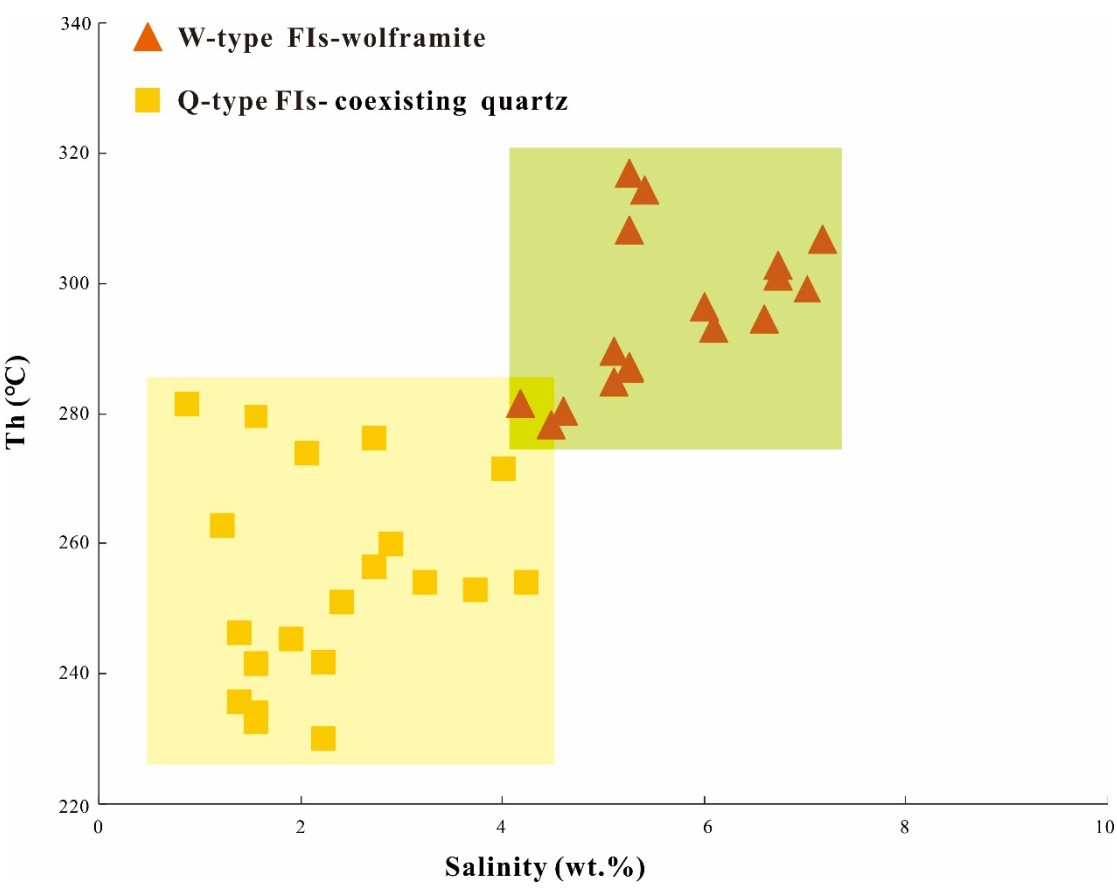

**Figure 16.** Homogenization temperature vs. salinity plot of inclusions in wolframite and its coexisting quartz.

Based on the above analysis and the $^{18}O$ values of quartz and wolframite, we can rule out boiling as a primary mineralization trigger (Figure 13). In the early stages of quartz veining (Figure 15I), meteoric water mixed with magmatic-hydrothermal fluid, causing cooling and a drop in ore metal solubility (and thus W ore precipitation). As a result, the temperature–salinity difference between wolframite and quartz fluids at Hongqiling is the result of wolframite precipitation preceding the coexisting quartz.

5.2.2. Role of $CO_2$ for Tungsten Mineralization

$CO_2$ is a common volatile component in W ore fluids and has played an important role in the formation of various types of deposits around the world [85]. Similarly, numerous studies in the Nanling region have found $CO_2$-rich fluid inclusions in other quartz-vein-type tungsten ores [86]. According to laser Raman spectroscopy, the quartz in the quartz-wolframite veins at Hongqiling contains $CO_2$ and $N_2$, but infrared microscopy revealed no $CO_2$-rich inclusions. This implies that the fluid composition of quartz cannot be used to predict the fluid processes of wolframite. According to [86], the earlier low-salinity $CO_2$-bearing aqueous solutions may represent the initial hydrothermal fluid before boiling. The low salinity of inclusions in wolframite-bearing quartz veins in our samples supports this hypothesis. After wolframite mineralization, boiling may begin in quartz vein mineralization. During boiling, $CO_2$ would be separated from the fluids, leaving a residual liquid of shallow $CO_2$ content to continue vein mineralization, which some L-type/V-type inclusions may represent in the S1 stage (Figure 15d–f). This can satisfactorily explain why $CO_2$ traces can be detected in both vapor and liquid phases of inclusions in wolframite-bearing quartz veins in laser Raman spectroscopy (Figure 9a,b).

As a result, while boiled fluid inclusions in quartz are uncommon, they may exist locally [86]. It must be acknowledged that, even if the mineralized fluid boils, the extent is most likely limited.

*5.3. Constraints on Quartz-Vein-Type Tungsten Ore Formation by Clastic Wall Rocks*

Previous research has suggested that different mineralization processes may be controlled by different wall rocks, affecting mineral assemblages and, ultimately, deposit types [14]. Shizhuyuan W-Sn polymetallic mineralization and Jinchuantang Sn-Bi polymetallic mineralization, for example, occurred in carbonate wall rocks. The Hongqiling quartz-vein-type Sn-W polymetallic mineralization, on the other hand, occurred in clastic wall rocks, demonstrating that different types of wall rocks have a direct influence on the mineralization process. We argue in this paper that the Hongqiling deposit's ore veins precipitation mechanism is a fluid mixing action of throughput and the resulting cooling. Furthermore, the different enclosing rocks that can influence the type of deposit also limit the mineralization mechanism of vein W ore.

Quartz-vein-type tungsten deposits are primarily found in the Caledonian (Sinian-Cambrian) folded basement area of South China [73]. The Hongqiling is the only quartz-vein-type Sn-W polymetallic deposit with Sinian clastic wall rocks in the Qianlishan-Qitianling area. It has distinct deposit genesis characteristics. Carbonate wall rock deposits (e.g., Shizhuyuan, Jinchuantang, Furong, and Huangshaping) were thought to have been boiled during the early magma intrusion and then mixed with low-temperature and low-salinity fluid in the middle and late ore stages. In contrast, the clastic rock-hosted Hongqiling ore formation likely did not undergo fluid boiling. Meanwhile, Xianghualing (another major W-Sn polymetallic orefield in the Nanling region) is consistent with this conjecture. Fluid boiling can be seen in the early stages of carbonate-hosted Nb-Ta-W-Sn mineralization. In contrast, the mineralization of W-Sn-Pb-Zn (hosted in the carbonate-clastic transition zone) occurred solely through fluid mixing.

It is widely assumed that when carbonate rocks mineralize enclosing rocks, metasomatic contact metamorphism occurs, resulting in skarn deposits. Because of their static chemical properties and brittle physical properties, clastic rocks are difficult to metasomatize and easily form fissures, resulting in vein-filled deposits. Second, in the original magma transport, carbonate rocks efficiently react with them. Furthermore, $CO_2$ exists as carbonate or bicarbonate ions in the hydrothermal fluid. In fluid transport, it produces immiscible or many inclusions containing daughter minerals such as rock salt and potassium salt, which provides a basis for confirming the occurrence of boiling. While clastic rocks are mineral-bearing wall rocks due to their numerous fissures, these fissures provide a channel for the transport and infiltration of other fluids, allowing mixing to occur. To summarize, the authors believe that the various types of enclosing rocks initially control the deposit type and deposit assemblage. Second, they may be the limiting factor for the vein tungsten ore fluid precipitation mechanism. This hypothesis is consistent with current research on vein Sn-W ores in the Nanling metallogenic belt. However, its applicability and breadth must be investigated and proven further.

## 6. Conclusions

(1) There are two types of fluid inclusions in the Hongqiling Sn-W polymetallic deposit: liquid-rich two-phase and vapor-rich two-phase. The mineralized fluid belongs to a medium-high temperature, low salinity NaCl-$H_2O$ fluid system. The broad range of fluid temperature suggests that the fluid activity associated with quartz formation is multistage. The ore-fluid evolution is characterized by decreasing temperature, salinity, and pressure, as well as a gradual increase in fluid density.

(2) Petrographic and microthermometric analyses reveal differences in the type, distribution, and size of inclusions between coexisting wolframite and quartz. W-type inclusions have higher homogenization temperatures and salinities than Q-type inclusions, indicating that wolframite was deposited before coexisting quartz. Its inclusions can better represent

early fluid evolution. The decreasing inclusion temperature, combined with the wolframite's narrow variation range, suggests that the wolframite precipitated via fluid mixing and simple cooling. No $CO_2$-bearing three-phase inclusions were found in the wolframite, but the presence of $CO_2$ was detected in the fluid inclusions in the coexisting quartz. We speculated that wolframite precipitation took place prior to fluid boiling. Furthermore, the early-vein low-salinity fluid represents the hydrothermal fluid before boiling occurred.

(3) The H-O isotope measurements revealed that magmatic and meteoric water mixed during the early ore stage (S1). Quartz from high-temperature veins may have formed first, then W-Sn mineralization may have developed along NE-trending structures. As meteoric water intrusion into the fluid system grew stronger, the ore fluid kept cooling. This could have led to extensive, intrusion-distal Pb-Zn-Ag mineralization by destabilizing and disintegrating the Pb-Zn complexes.

**Author Contributions:** Curation, J.X.; Investigation, H.H.; Methodology, E.Z.; writing—original draft preparation, W.R.; Writing—Review&Editing, L.W. and S.G. All authors have read and agreed to the published version of the manuscript.

**Funding:** This research was funded by National Key Research Development Program (2018YFC0603902); Yunnan Xingdian Talent Support Program 'Young Talents' Special Project.

**Data Availability Statement:** All data generated or used during the study appear in the submitted article.

**Acknowledgments:** The authors give special thanks to Chen Fuchuan from Kunming University of Science and Technology for his constructive reviews. We are deeply grateful to the leader and engineers in the Hongqiling Sn-W polymetallic deposit and Zhu Enyi, He Hao, and Li Bin from Kunming University of Science and Technology for their helpful field assistance.

**Conflicts of Interest:** The authors declare no conflict of interest.

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
