# Peer review of "Genesis and Fluid Evolution of the Hongqiling Sn-W Polymetallic Deposit in Hunan, South China: Constraints from Geology, Fluid Inclusion, and Stable Isotopes"

_minerals, doi:10.3390/min13030395_

Round 1

Reviewer 1 Report

Based on the geological, fluid inclusion and stable isotopical characteristics, the manuscript provides new microthermometry data, H-O and in-situ S isotopic compositions of the sulfides in the Hongqiling Sn-W polymetallic deposit. This study is helpful to further understand the metallogenesis of Sn-W deposits in the Nanling metallogenic belt, Southern China. This result is new and meaningful, worth publishing in Minerals. However, a few descriptions and discussion remain questionable. Overall this is a well-prepared manuscript and I suggest a minor revision. my detailed comments and suggestions are listed below.

1)      In Chapter 5.1.1, the authors concluded that the sulfur of the Hongqiling deposit was mainly from the Qianlishan granitic magma based on the δ34S value (-5.91~1.84‰). However, the author only provides data of 10 in situ S, is it persuasive? Consider increasing the amount of data appropriately to enhance persuasiveness. This will be helpful for the readers to accept the genetic assumptions of Hongqiling deposit.

2)      In the process of writing the article, you should identify the scientific problems to be solved in this study before stating what you plan to do: why do you want to do this study? What have been done? What problems remain? Maybe you can explain it in more detail in the "Introduction" section.

3)      Some of the references cited in this study is relatively old. Try to cite some latest ones.

4)       “All these indicate that the mineralizing fluid was derived from the granitic magma at Qianlishan”: the Qianlishan pluton is big, can you tell us which phase of the granite is related to the mineralization?

5)      How about the age determination of this deposit? As I know the age of this deposit is under debate. You may also discuss this point in this manuscript.

6)      Fig.1 b, the lower right corner small figure is too small to be read. Legends word in this figure is also too small to be read.

Huan Li

Author Response

Reviewer 1:

  • In Chapter 5.1.1, the authors concluded that the sulfur of the Hongqiling deposit was mainly from the Qianlishan granitic magma based on the δ34S value (-5.91~1.84‰). However, the author only provides data of 10 in situ S, is it persuasive? Consider increasing the amount of data appropriately to enhance persuasiveness. This will be helpful for the readers to accept the genetic assumptions of the Hongqiling deposit.

Response 1: appreciate the reviewer for his input regarding the possibility of adding data. Although I am already performing the in-situ S isotope additional test, I'm afraid that the data supplement will have to temporarily be postponed due to the requirement to adhere to the journal's change time limit. I think the current data can be explained, but supplementary explanation may be a better choice. However, it will take roughly two weeks to finish the test if the editorial department determines that additional data is required.

  • In the process of writing the article, you should identify the scientific problems to be solved in this study before stating what you plan to do: why do you want to do this study? What have been done? What problems remain? Maybe you can explain it in more detail in the "Introduction" section.

Response 2: thank the reviewer for his comments on the detailed introduction. This part of the manuscript has been revised.

  • Some of the references cited in this study is relatively old. Try to cite some latest ones.

Response 3: thank the reviewer for his comments about citing the latest references. New research achievements in recent years have been added in the discussion section.

  • “All these indicate that the mineralizing fluid was derived from the granitic magma at Qianlishan”: the Qianlishan pluton is big, can you tell us which phase of the granite is related to the mineralization?

Response 4: the Qianlishan granite stock is a complex pluton composed of three stages of rocks. Mao et al. (1995) divided this intrusive complex into three generations: (1) porphyritic biotite granite (152 ± 9 Ma) in the south; (2) equigranular biotite granite that formed the central part of the pluton (137 ± 7 Ma and 136 ± 6 Ma); (3) post-ore granite porphyry (131 ± 1 Ma). Fieldwork has confirmed that the Hongqiling area belongs to the second stage of the Qianlishan complex pluton. Its blind pluton has not been exposed for a long time, but our research group discovered in August 2022 during fieldwork that its pluton is now exposed. It is determined that the rock thin section is equigranular biotite granite after petrography work (As shown in the figure below, a,b: equigranular biotite granite).

  • How about the age determination of this deposit? As I know the age of this deposit is under debate. You may also discuss this point in this manuscript.

Response 5: Currently, our research team has two sets of data,158.9±0.7 and 158.4±0.8Ma from cassiterite in-situ U-Pb isochron age dating. The research team discussed whether to include the data in this study, but ultimately opted against it because it is not yet available for publication.

  • 1 b, the lower right corner small figure is too small to be read. The Legends word in this figure is also too small to be read.

Response 6: Figure 1 has been revised.

Reviewer 2 Report

I reviewed the manuscript "Genesis and fluid evolution of the Hongqiling Sn-W polymetallic deposit in Hunan, South China: Constraints from geology, fluid inclusion, and stable isotopes".

The authors reported interesting conclusions on the genesis of the Sn-W 2 polymetallic deposit in Hunan (South China) based on studies of fluid inclusions and complex isotopic characteristics. In my opinion, the results are very interesting and can be published in the journal after correction of few remarks.

Overall, the manuscript is written in a good scientific language, well illustrated and logical. The Introduction provides a brief overview of the current state of the problem, the main scientific question and the problems that the authors are solving in the process of their research.

The Sampling and Methodology section needs to be improved. All analytical methods are thoroughly described, but there is a lack of specifics about the samples. My wish is to make a table with GPS location of samples and a brief description of which veins belong to which samples.

The Results are presented in a structured and comprehensive style. The abundance of figures makes this section (as well as the Discussion) understandable and very clear. The Discussion section reviews the main results of the work and provides the main conclusions of the study. The Conclusion follows logically from the Discussion. The main conclusions and results of the study are presented in the form of short thesis statements, which is quite acceptable.

My remarks:

1. Figure 1 and 2. The small fonts and individual elements of these figures are difficult to recognize. If you can, try to enlarge the fonts and details in the figures.

2. Figure 11.  Try to correct the small fonts.

3.   All tables in the manuscript should be made according to the requirements of the journal.

4. The formatting of the manuscript should be improved according to the requirements of the journal.

5.   Carefully check all references to literature.

Author Response

Reviewer 2:

  1. Figure 1 and 2. The small fonts and individual elements of these figures are difficult to recognize. If you can, try to enlarge the fonts and details in the figures.

Response 1: Thank the reviewer for his/her suggestions. Figure 1 has been revised.

  1. Figure 11.  Try to correct the small fonts.

Response 2: Figure 11 has been revised.

  1. All tables in the manuscript should be made according to the requirements of the journal.

Response 3: Tables have been checked and revised.

  1. The formatting of the manuscript should be improved according to the requirements of the journal.

Response 4: Thank the editor of the editorial department for helping me modify the format, the formatting of the manuscript has been revised.

  1. Carefully check all references to literature.

Response 5: The references have been carefully rechecked.

Reviewer 3 Report

Line 9 - "Correspondence" - duplicated, correct it.

Line 11 - It is (not "And is")

Line 12 - "alteration"? - What alteration stages do you have in the deposit?

Line 13 and throughout the text - "wolframite" - In fact, this is not a mineral species classified by the IMA, but a group name of mineral series with Huebnerite and Ferberite end members. Do you have any data on the mineral species affiliation (Fe-rich or Mn-rich) of “wolframite" you described? If no, explain in the text that you described an unspecified member of this series.

Line 43 - "Hongqiling Sn-polymetallic deposit" or "Hongqiling Sn-W polymetallic deposit" (as in the title)

Line 95 - Why 1995a? Do you have Mao et al. (1995b)? Fix it.

Line 118-119 - "Hongqiling pluton"? - Do you have Hongqiling pluton? Is it different from the Qianlishan pluton? Where is it on the map?

Line 142-147 - Why (+) is needed, just list the minerals identified: cassiterite, wolframite, scheelite, arsenopyrite, etc.

Table 1:

- Correct some captions and terms in the table, e.g.: Orebody number, Ore type, Ore texture, metasomatiC (not metasomatiE)

- No need to repeat "texture", "structure" and "type" in the columns of the same name, just: cassiterite-sulfide, Pb-Zn, granular, cataclastic, massive, disseminated, banded, vein, etc.

Fig. 3:

- Better: Mineralization stage (not "Mettalogenic period/stage")

- Do you really have two different substages (W(Sn) and Sn(W)) in S1? How did you distinguish them? Explain this in the text (Line 142 and thereafter).

Fig. 4-5:

- Use lowercase (as in other Figs) for Fig. 5 both on the photos and in the captions.

- List the mineral abbreviations used on the photos (no matter they are understandable from the description).

- Label the chalcopyrite (Cp) on the Fig. 5a.

Line 212 - Better: H–O isotope analysis (you analyzed not only quartz but also wolframite)

Line 236 - Fluid inclusionS

Line 238 - "Fluid" - lowercase

Line 278 - in quartz? Write it.

Line 303 - "in the liquid phase"? - Fig. 9a is for the vapor phase!

Line 304 - "in the vapor phase decreases"? - CO2 decreases in the liquid phase (Fig. 9b)!

Line 331 - "P = 105 Pa"? What is P in this equation?

Line 331, 336 - (×105 Pa)

Line 332-334 - How these ore fluid pressure values correspond to those in Table 2? Correct them!

Line 337-339 - The same: how these values correspond to those given in Table 2? Correct them!

Line 345 - "(1.1-14.7)" - 1.1 not corresponding to the data in Table 3, correct it!

Table 3 and 4 - Include the cited references (Xiao, 1989; Chen et al., 1992) in the Reference List and also use the Reference number when cited.

Line 370 - Better: Ore-forming fluid source and evolution

Fig. 13 - Mark labels (a, b) on the plots. Explain the red and blue points and rectangles.

Line 461 - "Faults" - lowercase

Line 465-475 - One sentence? Edit it!

Fig. 14b - What is the meaning of the numbers on the plot? Explain them in the figure caption.

Line 531 - "Sedimentation"? – better: Deposition

Line 585 - the authorS believe

Line 619 - ReferenceS

Author Response

Reviewer 3:

Line 9 - "Correspondence" - duplicated, correct it.

revised

Line 11 - It is (not "And is")

revised

Line 12 - "alteration"? - What alteration stages do you have in the deposit?

Has been changed to mineralization stages.

Line 13 and throughout the text - "wolframite" - In fact, this is not a mineral species classified by the IMA, but a group name of mineral series with Huebnerite and Ferberite end members. Do you have any data on the mineral species affiliation (Fe-rich or Mn-rich) of “wolframite" you described? If no, explain in the text that you described an unspecified member of this series.

Thank the reviewer for his/her rigorous suggestions. In fact, I have classified the wolframite in the study area. Through the EPMA study, it is concluded that the wolframite in Hongqiling mineral has the characteristics of Fe-rich. However, because this data is an unpublished research result, it cannot be explained in this article.

Line 43 - "Hongqiling Sn-polymetallic deposit" or "Hongqiling Sn-W polymetallic deposit" (as in the title)

revised

Line 95 - Why 1995a? Do you have Mao et al. (1995b)? Fix it.

revised

Line 118-119 - "Hongqiling pluton"? - Do you have Hongqiling pluton? Is it different from the Qianlishan pluton? Where is it on the map?

Due to the misunderstanding caused by my imprecise statement, the Hongqiling pluton is a deep blind pluton, and according to previous studies, in fact, the Hongqiling pluton is the continuation of the Qianlishan pluton in the deep.

Line 142-147 - Why (+) is needed, just list the minerals identified: cassiterite, wolframite, scheelite, arsenopyrite, etc.

revised

Table 1:

- Correct some captions and terms in the table, e.g.: Orebody number, Ore type, Ore texture, metasomatiC (not metasomatiE)

- No need to repeat "texture", "structure" and "type" in the columns of the same name, just: cassiterite-sulfide, Pb-Zn, granular, cataclastic, massive, disseminated, banded, vein, etc.

revised

Fig. 3:

- Better: Mineralization stage (not "Mettalogenic period/stage")

- Do you really have two different substages (W(Sn) and Sn(W)) in S1? How did you distinguish them? Explain this in the text (Line 142 and thereafter).

Thank the reviewer for his/her rigorous comment. After consideration, two different substages have been deleted. In fact, they cannot be clearly divided. Figure 3 has been revised.

Fig. 4-5:

- Use lowercase (as in other Figs) for Fig. 5 both on the photos and in the captions.

- List the mineral abbreviations used on the photos (no matter they are understandable from the description).

- Label the chalcopyrite (Cp) on the Fig. 5a.

revised

Line 212 - Better: H–O isotope analysis (you analyzed not only quartz but also wolframite)

revised

Line 236 - Fluid inclusionS

revised

Line 238 - "Fluid" – lowercase

revised

Line 278 - in quartz? Write it.

revised

Line 303 - "in the liquid phase"? - Fig. 9a is for the vapor phase!

revised

Line 304 - "in the vapor phase decreases"? - CO2 decreases in the liquid phase (Fig. 9b)!

revised

Line 331 - "P = 105 Pa"? What is P in this equation?

Errors caused by carelessness, revised.

Line 331, 336 - (×105 Pa)

revised

Line 332-334 - How these ore fluid pressure values correspond to those in Table 2? Correct them!

revised

Line 337-339 - The same: how these values correspond to those given in Table 2? Correct them!

revised

Line 345 - "(1.1-14.7)" - 1.1 not corresponding to the data in Table 3, correct it!

revised

Table 3 and 4 - Include the cited references (Xiao, 1989; Chen et al., 1992) in the Reference List and also use the Reference number when cited.

References have been added.

Line 370 - Better: Ore-forming fluid source and evolution

revised

Fig. 13 - Mark labels (a, b) on the plots. Explain the red and blue points and rectangles.

revised

Line 461 - "Faults" – lowercase

revised

Line 465-475 - One sentence? Edit it!

revised

Fig. 14b - What is the meaning of the numbers on the plot? Explain them in the figure caption.

Has been explained in the figure caption title.

Line 531 - "Sedimentation"? – better: Deposition

revised

Line 585 - the authorS believe

revised

Line 619 - ReferenceS

revised

All revisions have been marked in yellow in the manuscript. Thank you again for all comments of the reviewer.
